# A common mechanism for recruiting the Rrm3 and RTEL1 accessory helicases to the eukaryotic replisome

Ottavia Olson[1], Simone Pelliciari[1,2], Emma D Heron[1,2] & Tom D Deegan [1✉]

## Abstract

The eukaryotic replisome is assembled around the CMG (CDC45-MCM-GINS) replicative helicase, which encircles the leading-strand DNA template at replication forks. When CMG stalls during DNA replication termination, or at barriers such as DNA-protein crosslinks on the leading strand template, a second helicase is deployed on the lagging strand template to support replisome progression. How these 'accessory' helicases are targeted to the replisome to mediate barrier bypass and replication termination remains unknown. Here, by combining AlphaFold structural modelling with experimental validation, we show that the budding yeast Rrm3 accessory helicase contains two Short Linear Interaction Motifs (SLIMs) in its disordered N-terminus, which interact with CMG and the leading-strand DNA polymerase Polε on one side of the replisome. This flexible tether positions Rrm3 adjacent to the lagging strand template on which it translocates, and is critical for replication termination in vitro and Rrm3 function in vivo. The primary accessory helicase in metazoa, RTEL1, is evolutionarily unrelated to Rrm3, but binds to CMG and Polε in an analogous manner, revealing a conserved docking mechanism for accessory helicases in the eukaryotic replisome.

Keywords Accessory Helicase; CMG Helicase; DNA Replication; Rrm3; RTEL1

Subject Categories DNA Replication, Recombination & Repair; Structural Biology

## Introduction

In all domains of life, genomic DNA is replicated by macromolecular machines called replisomes, which are generally assembled around ring-shaped hexameric replicative helicases (O'Donnell and Li, 2018). The eukaryotic replicative helicase is formed by six related Mcm2-7 AAA+ ATPases, which associate with CDC45 and the GINS tetramer to form the CDC45-MCM-GINS (CMG) complex (Costa and Diffley, 2022). The Mcm2-7 catalytic motor of CMG encircles and translocates upon the template strand for leading strand synthesis at replication forks, whilst simultaneously excluding the lagging strand template from its central channel, thereby unwinding DNA via a classic steric exclusion mechanism (Fu et al, 2011).

By topologically entrapping the leading strand template, CMG can remain stably associated with replication forks, driving processive unwinding over many hundreds of kbp in mammalian cells (Mechali, 2010). Despite this, CMG-driven unwinding is frequently challenged during normal DNA replication elongation (Shyian and Shore, 2021). CMG can stall, for example, at naturally occurring barriers formed by stably bound proteins, such as those found at centromeres, telomeres, inactive DNA replication origins, and heavily transcribed genes (Claussin et al, 2022; Ivessa et al, 2003; Ivessa et al, 2002; Osmundson et al, 2017; Tran et al, 2017). CMG is also impeded by covalent DNA-protein crosslinks (DPCs) (Duxin et al, 2014; Sparks et al, 2019), which can form as a result of aberrant topoisomerase activity, or in response to chemotherapeutic drugs and other chemical agents. Such barriers represent particularly penetrant blocks to unwinding when they are positioned on the leading strand template, upon which CMG translocates (Duxin et al, 2014; Sparks et al, 2019). Finally, we showed previously that CMG also stalls during DNA replication termination, when two replisomes emanating from neighbouring replication origins converge upon one another (Deegan et al, 2019).

The predominant mechanism that supports replisome progression past CMG-blocking barriers (and during replication termination) involves the deployment of a 2nd so-called 'accessory' DNA helicase at replication forks (Bruning et al, 2014; Shyian and Shore, 2021). The archetypal eukaryotic accessory helicases are budding yeast Rrm3 and Pif1, which both belong to Pif1-family of 5′-3′ DNA helicases (Bochman et al, 2010). Of these two paralogues, Rrm3 is the more important upon CMG stalling, with Pif1 playing a back-up role at some CMG-blocking barriers, and during replication termination (Deegan et al, 2019; Ivessa et al, 2003; Ivessa et al, 2002; Osmundson et al, 2017; Tran et al, 2017). Accordingly, cells lacking Rrm3 exhibit markedly increased replisome stalling at the promoters of heavily transcribed tRNA genes, centromeres, dormant replication origins, programmed Fob1 barriers in the ribosomal DNA (rDNA) repeats, and at sites of

[1]MRC Human Genetics Unit, Institute of Genetics and Cancer, University of Edinburgh, Western General Hospital, Edinburgh EH4 2XU, UK. [2]These authors contributed equally: Simone Pelliciari, Emma D Heron. ✉E-mail: tdeegan@ed.ac.uk

replication termination (Claussin et al, 2022; Deegan et al, 2019; Ivessa et al, 2003; Ivessa et al, 2002; Osmundson et al, 2017; Tran et al, 2017).

Pif1-family helicases are present across evolution (Bochman et al, 2010), but the function of PIF1 in metazoan DNA replication is less well-established than for budding yeast Rrm3 and Pif1 (Snow et al, 2007). Metazoa also contain a number of other 5′-3′ helicases, of which HELB, DDX11, FANCJ and RTEL1 have all been ascribed a role in some aspect of chromosome replication (Campos et al, 2023; Hazeslip et al, 2020; Jegadesan and Branzei, 2021; Lerner et al, 2020; Sparks et al, 2019; Vannier et al, 2013; Yaneva et al, 2023). The RAD3-related DNA helicase RTEL1, mutations in which cause the human genetic diseases Dyskeratosis Congenita and Hoyeraal-Hreidarsson syndrome (Vannier et al, 2014), was previously described as a functional analogue of the yeast anti-recombinase Srs2 (Barber et al, 2008). However, subsequent pioneering work in *Xenopus* egg extracts has shown that RTEL1 is also required for replisome progression past leading strand DPCs, stably bound proteins, and during replication termination (Campos et al, 2023; Sparks et al, 2019). Taken together with earlier observations that RTEL1 is required for normal replication fork progression in mouse embryonic fibroblasts (Vannier et al, 2013), these data suggest that metazoan RTEL1 might be a functional analogue of budding yeast Rrm3 in the eukaryotic replisome.

A universal feature of the aforementioned eukaryotic accessory helicases is that they translocate with 5′-3′ polarity. Thus, it is generally assumed that these helicases support termination and the bypass of barriers by loading onto the lagging strand template and unwinding beyond the blocked CMG helicase on the leading strand (Bruning et al, 2014; Shyian and Shore, 2021). In support of this model, a barrier on the lagging strand template impedes RTEL1 function during replisome bypass of a leading strand DPC in *Xenopus* egg extracts (Sparks et al, 2019). In the case of DPC bypass, once the accessory helicase has generated a sufficient amount of single-stranded DNA (ssDNA) beyond the stalled CMG, CMG can 'traverse' past the DPC by an as yet undetermined mechanism, and DNA replication can then resume (Sparks et al, 2019). However, at other barriers, such as Mcm2-7 double hexamers at dormant replication origins (Hill et al, 2020), the blocking proteins are disassembled from the DNA as a result of accessory helicase activity on the lagging strand template.

Notably, in prokaryotes such as *E. coli*, where the replicative helicase DnaB translocates 5′-3′ on the lagging strand template, the 3′-5′ helicases Rep and UvrD are required to support replisome progression past DnaB-blocking protein barriers (Bruning et al, 2014; Guy et al, 2009). Thus, the use of an accessory helicase with the opposite polarity to the replicative helicase appears to be an evolutionarily conserved feature of replisome organisation. However, whilst structural information is available for some eukaryotic accessory helicases in isolation (Dehghani-Tafti et al, 2019; Su et al, 2019), there is very little currently known about their positioning and structure in the context of the replisome. Furthermore, whether specific recruitment mechanisms exist to target eukaryotic accessory helicases to the lagging strand template remains unknown. It is also unclear whether the different accessory helicases present in yeasts and metazoa share any common features that are important for their function in DNA replication. Here, we describe a mechanism, based on direct interactions with the CMG replicative helicase and DNA polymerase ε, for recruiting budding yeast Rrm3 and metazoan RTEL1 to the eukaryotic replisome.

## Results

### The Rrm3 IDR is required for Rrm3 function

Our previous work showed that a budding yeast Pif1-family helicase (Rrm3 or Pif1) is required to drive efficient DNA replication termination in the origin-dependent budding yeast in vitro DNA replication system (Deegan et al, 2019). Whilst either Rrm3 or Pif1 can fulfil this role in vitro, Rrm3 is more important during replication termination and replisome progression past protein barriers in vivo (Claussin et al, 2022; Deegan et al, 2019; Ivessa et al, 2003; Ivessa et al, 2002; Osmundson et al, 2017; Tran et al, 2017). Thus, we focussed on Rrm3. Our previous work also showed that a bacterial Pif1-family DNA helicase (BacPif1), which is related to budding yeast Pif1 and Rrm3, cannot support DNA replication termination in vitro (Deegan et al, 2019). Structural comparison indicated a high degree of conservation between the helicase domains of Rrm3 and BacPif1 (Fig. 1A). However, consistent with computational predictors of protein disorder (Fig. EV1A), AlphaFold2 (Jumper et al, 2021) predicts that Rrm3 contains an ~230 residue Intrinsically Disordered Region (IDR) at its N-terminus, which is absent from BacPif1 (Fig. 1A).

Guided partly by previous genetic experiments involving a series of *rrm3* truncation mutants (Bessler and Zakian, 2004), we initially sought to assess the functional importance of the Rrm3 IDR by testing whether a version of Rrm3 that lacked the first 193 residues of the predicted IDR (Rrm3ΔN) could support replication termination in the origin-dependent budding yeast in vitro DNA replication system. To specifically monitor termination efficiency, nascent replication products are linearised by restriction digest and then resolved on native agarose gels, such that any fully replicated molecules migrate as full-length linear species. If converging replisomes stall as they approach each other during termination, then replicated plasmids are still linked by a short stretch of parental DNA, and migrate as large Late Replication Intermediates (LRIs). Whilst full-length Rrm3 routinely stimulated termination in vitro by 4-5-fold, Rrm3ΔN could not support termination, as evidenced by the persistence of LRIs (Fig. 1B,C). Importantly, Rrm3ΔN showed a similar low level of activity as full-length Rrm3 in a simple helicase assay (Fig. EV1B,C), indicating that the Rrm3 IDR, which is absent from BacPif1, is required for some other aspect of Rrm3 function during replication termination in vitro.

Rrm3 is essential in budding yeast that lack the Dia2 subunit of the SCF^Dia2 E3 ubiquitin ligase (Fig. EV1D) (Morohashi et al, 2009). This reflects an essential requirement for Rrm3 to remove 'old' terminated CMG (bound around double-stranded DNA) from the path of progressing replisomes, when CMG has not been disassembled by the cognate SCF^Dia2-driven pathway in the previous cell cycle (Polo Rivera et al, 2024). Cells lacking Rrm3 also exhibit an increased reliance on the apical DNA replication checkpoint kinase Mec1 for cell growth (Fig. EV1E) (Ivessa et al, 2003), likely due to enhanced replication fork stalling in the absence of Rrm3 (Claussin et al, 2022; Deegan et al, 2019; Ivessa et al, 2003; Ivessa et al, 2002; Osmundson et al, 2017; Tran et al, 2017). Both of these

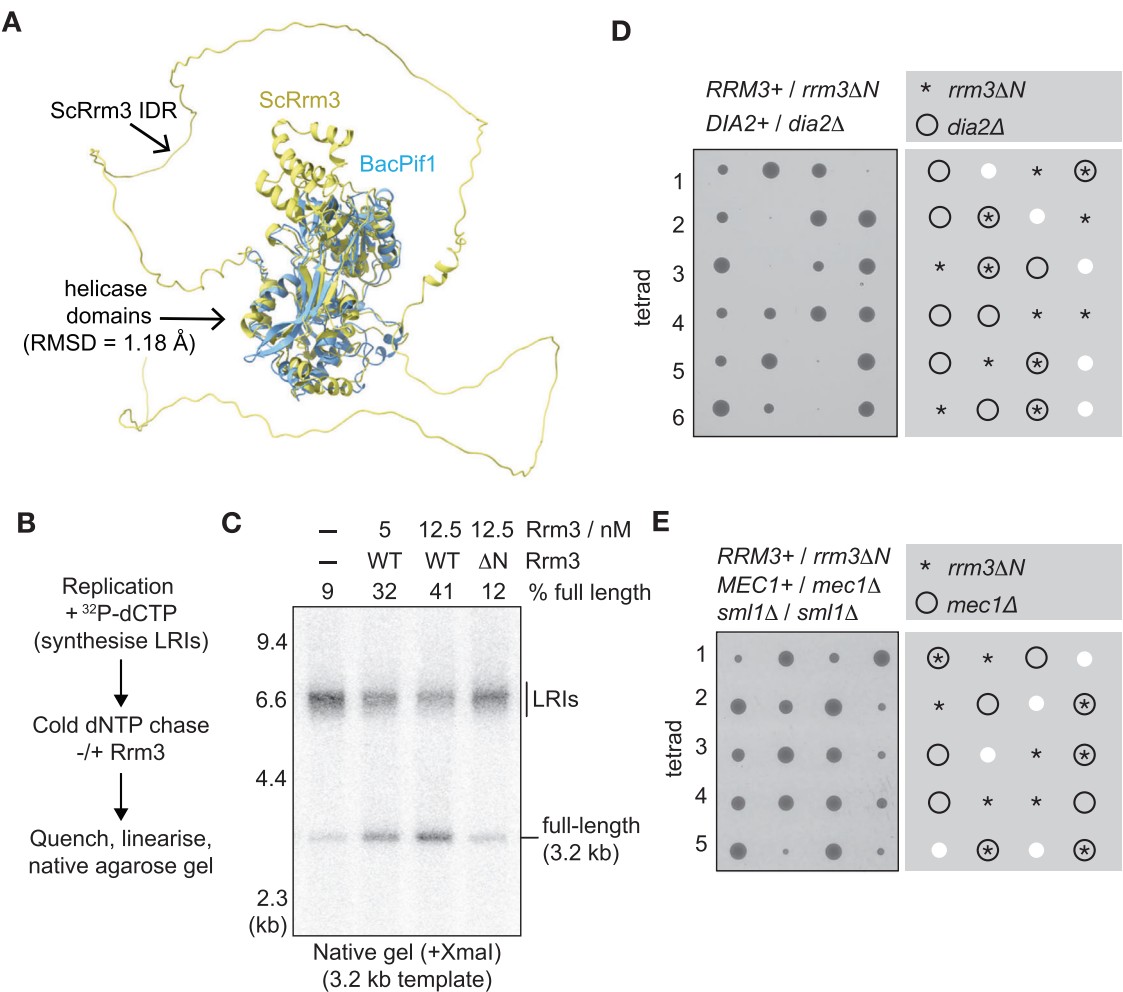

**Figure 1. The N-terminal IDR of Rrm3 is required for Rrm3 function.**

(A) Comparison of AlphaFold2-predicted structures of *S. cerevisiae* Rrm3 with Pif1 from *Bacteroides* sp 2 1 16 (BacPif1), showing that the Rrm3 Intrinsically Disordered Region (IDR) is absent from BacPif1. Root mean square deviation (RMSD) is given for the aligned helicase domains. (B) Reaction scheme for examining DNA replication termination in in vitro DNA replication reactions reconstituted with purified *S. cerevisiae* proteins. LRIs = Late Replication Intermediates, which result from the stalling of converging replisomes in the absence of *S. cerevisiae* Rrm3 or Pif1. (C) A 3189 bp plasmid template (pBS/ARS1WTA) was replicated according to the scheme in (B), with wild-type (WT) Rrm3 or Rrm3 lacking the first 193 amino acids (ΔN) added as indicated. XmaI-digested radiolabelled replication products were resolved in a native agarose gel and detected by autoradiography. (D, E) Diploid yeast cells of the indicated genotypes were sporulated and the resulting tetrads were then dissected and grown on YPD medium for 2 days at 30 °C. Source data are available online for this figure.

synthetic genetic relationships were recapitulated with the *rrm3ΔN* allele (Fig. 1D,E), indicating that the Rrm3 IDR is required for Rrm3 function in vivo, consistent with previously published data (Bessler and Zakian, 2004).

## The Rrm3 IDR mediates Rrm3 binding to CMGE

We hypothesised that the N-terminal IDR of Rrm3 might mediate Rrm3 recruitment to the replisome. Neither the polymerase sliding clamp PCNA, its loader RFC, nor the accessory replisome proteins Ctf4, Mrc1 or Tof1-Csm3 are required for Rrm3-dependent termination in the reconstituted DNA replication system (Fig. EV2A), suggesting that CMG and Polε (which together can form a minimal leading strand replisome complex called CMGE (Langston et al, 2014)) might be sufficient to support Rrm3 function at replication forks. Thus, we sought to test whether the Rrm3 IDR

interacts with CMGE. We incubated purified Rrm3, CMG and Polε together and isolated resultant complexes via immunoprecipitation of the Sld5 subunit of CMG (Fig. 2A). Rrm3 interacted with CMGE, but this interaction was completely abolished with Rrm3ΔN (even when using 10-fold higher protein concentrations than for Rrm3), indicating that the IDR is required for Rrm3 binding to CMGE (Fig. 2A).

To test if the Rrm3 IDR is sufficient for CMGE binding, we purified a chimeric protein (Rrm3N-BacPif1) in which the first 193 residues of Rrm3 were fused to N-terminal end of the BacPif1 helicase domain (Fig. 2B). This fusion protein showed comparable helicase activity to BacPif1 (Figs. 2C and EV2), but was able to interact with budding yeast CMGE (Fig. 2D), whereas BacPif1 was not (Fig. 2E). Strikingly, this ability to bind to budding yeast CMGE correlated with an ~4–6-fold increase in the capacity of Rrm3N-BacPif1 to support termination in the budding yeast replication

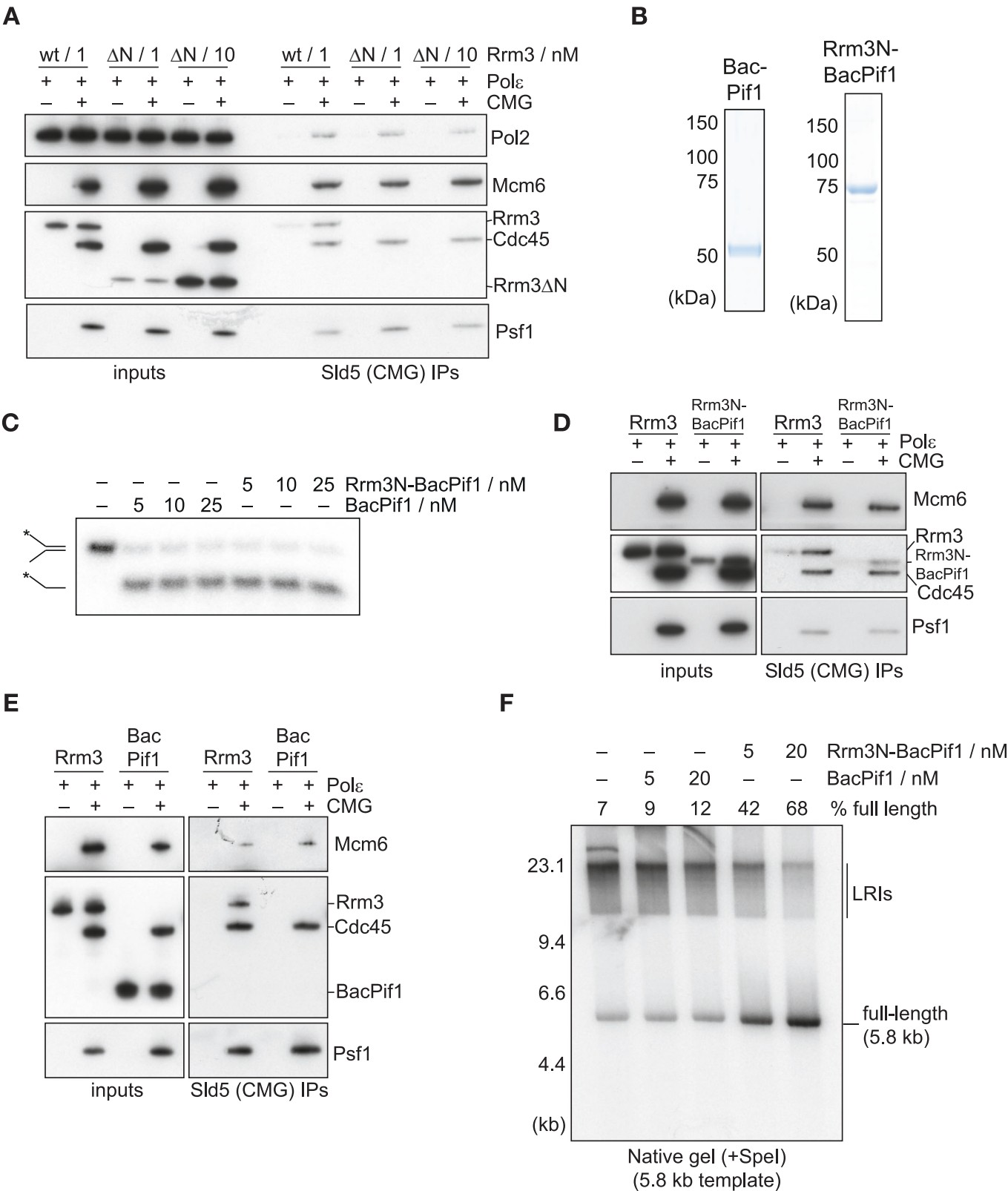

◀ **Figure 2. The Rrm3 IDR is necessary and sufficient for CMGE binding.**

(A) The ability of Rrm3 and Rrm3ΔN to associate with CMG was monitored in the presence of Polε. The indicated factors were mixed, before immunoprecipitation of the Sld5 subunit of CMG and immunoblotting. Rrm3 and Cdc45 were detected by anti-FLAG immunoblotting in this and subsequent experiments. (B) Purified BacPif1 and a version of BacPif1 that was fused to the first 193 amino acids of the Rrm3 IDR (Rrm3N-BacPif1) visualised by SDS-PAGE and Coomassie staining. (C) The ability of BacPif1 and Rrm3N-BacPif1 to unwind a 25 bp DNA duplex, formed by annealing oligonucleotides TD254 and TD255, was monitored as described in Methods. * indicates $^{32}$P-labelling of TD254. (D, E) The ability of Rrm3, Rrm3N-BacPif1 and BacPif1 to associate with CMG was monitored in the presence of Polε as in (A). (F) In vitro DNA replication reactions conducted as in Fig. 1B with the indicated concentrations of Rrm3N-BacPif1 and BacPif1. Source data are available online for this figure.

system, compared with BacPif1 (Figs. 2F and EV2C). Taken together, these data indicate that the Rrm3 IDR is necessary and sufficient for Rrm3 binding to CMGE. Furthermore, these data indicate that the inability of BacPif1 to support replication termination in vitro (Deegan et al, 2019) and complement deletion of *RRM3* in vivo (Andis et al, 2018) is very likely due to a failure to be efficiently recruited to the budding yeast replisome.

## Molecular mechanism of Rrm3 binding to CMGE

To understand the molecular mechanism of Rrm3 IDR binding to CMGE, we used AlphaFold-Multimer (Evans et al, 2022) to perform an in silico screen for interactions between Rrm3 and core budding yeast replisome proteins (Appendix Table S1). AlphaFold-Multimer did not predict interactions between Rrm3 and Ctf4, Mrc1 or Tof1-Csm3, consistent with Ctf4, Mrc1 and Tof1-Csm3 not being required to support Rrm3 function during replication termination in vitro (Fig. EV2A). In two out of the five models generated, AlphaFold-Multimer did predict an interaction between PCNA and a previously characterised PIP box in Rrm3 residues 35–42 (Appendix Table S1) (Schmidt et al, 2002). However, PCNA is completely dispensable for Rrm3-driven termination in vitro (Fig. EV2A), consistent with previous data showing that Rrm3 PIP box mutations do not compromise Rrm3 function in vivo (Syed et al, 2016; Varon et al, 2024). Thus, we focussed on the high confidence pairwise interactions that were predicted for the Rrm3 IDR with two components of CMGE (Appendix Table S1): Sld5, a subunit of the GINS complex, and Dpb2, the second largest subunit of Polε (Fig. 3A,B and Appendix Fig. S1).

The predicted interactions with Dpb2 and Sld5 are mediated by adjacent Short Linear Interaction Motifs (SLIMs) in the Rrm3 IDR (Fig. 3C). The Rrm3-Dpb2 interaction involves Rrm3 residues 86–110, which are predicted to form a continuous ~50 Å interface along an L-shaped cleft on the surface of Dpb2 (Fig. 3A,C). The Rrm3-Sld5 interface is smaller, and is mediated primarily by Rrm3 residues 114–122, which are predicted to bind a surface patch of mixed hydrophobic and acidic nature on Sld5 (Fig. 3B,C).

Docking of the Rrm3-Sld5 and Rrm3-Dpb2 AlphaFold-Multimer models onto a structure of the budding yeast replisome (Jenkyn-Bedford et al, 2021) revealed two important features of the predicted Rrm3-CMGE interface (Fig. 3D). Firstly, the two predicted Rrm3 interaction sites form an almost continuous surface on one side of CMGE. This arrangement is complementary to the close spacing of the two adjacent SLIMs in the Rrm3 IDR (Fig. 3C), and very likely permits simultaneous engagement of these two neighbouring SLIMs onto a single Rrm3 docking site, formed by both Dpb2 and Sld5. Secondly, the predicted Rrm3 docking site is adjacent to where the lagging strand DNA template emerges from the central channel of CMG (Jenkyn-Bedford et al, 2021), between the Mcm3 and Mcm5 subunits, revealing a compelling mechanism for localising the

Rrm3 helicase domain (connected to the CMGE binding SLIMs in Rrm3 by ~100 amino acids of disordered polypeptide) close to the lagging strand template on which it translocates (Fig. 3D).

## Isolation of CMGE-binding mutants in Rrm3

To test the AlphaFold-Multimer predictions, we generated a range of Rrm3 mutant proteins, designed to specifically disrupt either the Rrm3-Sld5 or Rrm3-Dpb2 interactions (Figs. 4A and EV3A). Wild-type FLAG-tagged Rrm3 interacted with the GINS tetramer via co-immunoprecipitation onto anti-FLAG beads (Fig. 4B). This interaction was disrupted by either deletion of the Sld5-binding SLIM (Rrm3Δ111-130), or mutation of two key positively charged residues to glutamate (Rrm3-2E) in this region (Fig. 4A,B). Likewise, wild-type Rrm3 interacted directly with Polε (Fig. 4C), as previously suggested by co-immunoprecipitation studies in yeast cell extracts (Azvolinsky et al, 2006). The Rrm3-Polε interaction was disrupted by mutation of key interacting residues in Rrm3 (Rrm3-6A and Rrm3-CR), and abolished by deletion of the entire Dpb2-binding SLIM (Rrm3Δ86-110) (Fig. 4A,C).

The Sld5- and Dpb2-binding mutants of Rrm3 were proficient for binding to either Polε or GINS (respectively) (Fig. EV3B,C), thus enabling us to specifically assess the contribution of each SLIM individually to Rrm3 recruitment and function. Rrm3 binding to reconstituted CMGE complexes was partly disrupted by mutation of the Sld5-binding SLIM (Rrm3-2E), and completely abolished by deletion of the Dpb2-binding SLIM (Rrm3Δ86-110) (Fig. 4D), indicating that the Sld5- and Dpb2-binding sites jointly contribute to Rrm3 recruitment to CMGE, with the larger Dpb2-binding site the more important under these experimental conditions.

## CMGE binding is critical for Rrm3 function in vitro and in vivo

We next investigated the importance of CMGE binding by Rrm3 for Rrm3-driven termination in reconstituted in vitro DNA replication reactions (Fig. 5A,B). Rrm3-2E and Rrm3Δ111-130 were proficient for LRI resolution (Figs. 5A and EV4A), indicating that the Rrm3-Sld5 interaction is dispensable for termination in vitro. Strikingly, however, mutation of the Dpb2-binding SLIM in Rrm3 reproducibly lead to a partial reduction in termination efficiency, whilst deletion of this SLIM completely abolished the capacity of Rrm3 to support termination in vitro (Figs. 5B and EV4B).

To assess the significance of the recruitment mechanism we identified for Rrm3 function in vivo, we introduced the *rrm3-2E, -6A, -CR* and *Δ86-110* mutations into a single copy of the *RRM3* gene, in diploid yeast strains that were heterozygous for deletion of *DIA2* (Figs. 5C,D and EV4C,D). Tetrad dissection of the resultant

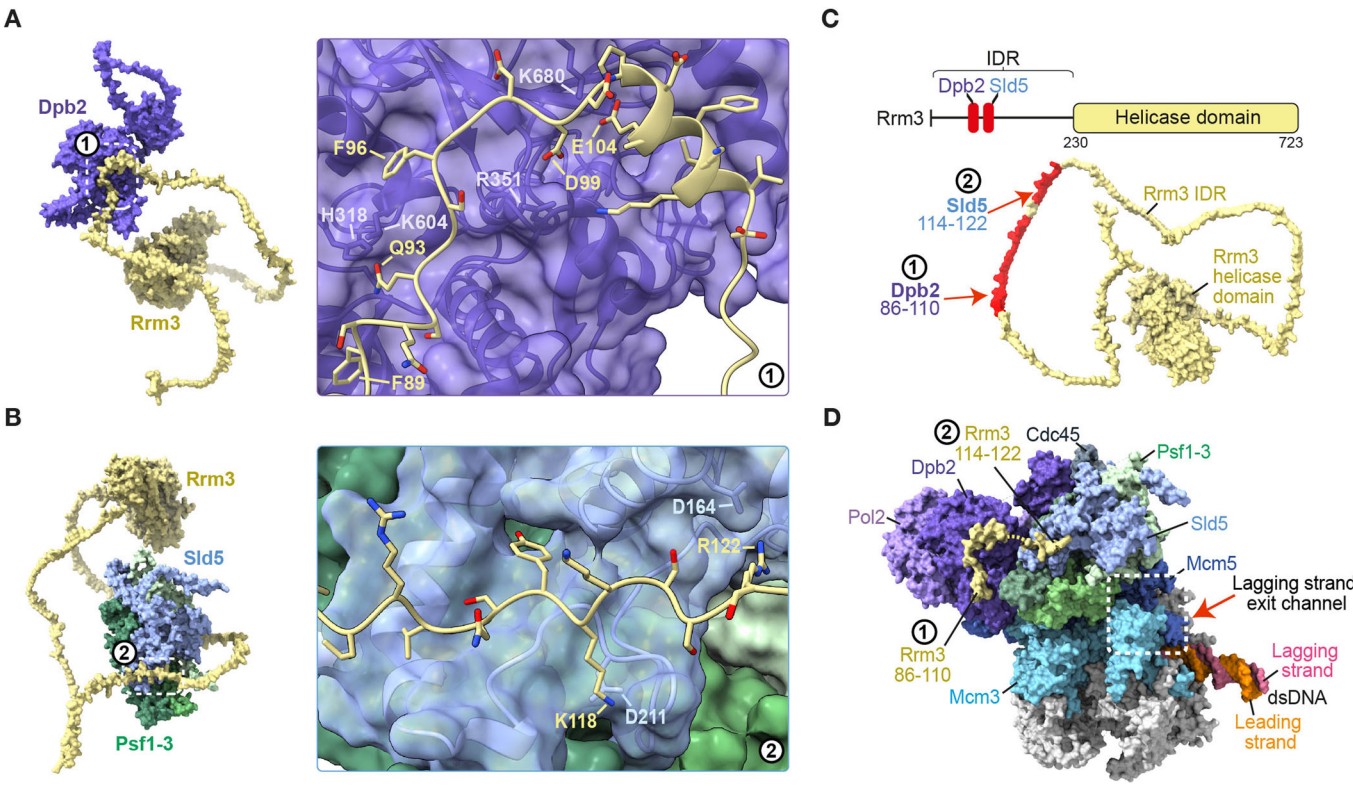

**Figure 3. Structural modelling of Rrm3 in the budding yeast replisome.**

(A, B) AlphaFold-Multimer models of Rrm3 bound to Dpb2 (A) and the GINS tetramer (B). Key interaction residues that were mutated in Rrm3 mutants (Fig. 4) are indicated. (C) Domain structure and AlphaFold2-predicted monomer structure of Rrm3. Positions of adjacent Dpb2- and Sld5-binding Short Linear Interaction Motifs (SLIMs) in the Rrm3 IDR are indicated. (D) Dpb2- and Sld5-binding SLIMs in Rrm3 were docked onto a cryo-EM structure of budding yeast CMG-Polε (PDB: 7PMK) by aligning on Dpb2 and Sld5, respectively (see Methods for more details). The disordered segment of Rrm3 that connects the two SLIMs is represented as a dashed line. The path of the excluded lagging strand DNA template, which exits the CMG central channel between Mcm3 and Mcm5, is indicated.

strains to generate haploid spores indicated that *dia2Δ* cells were very sick in combination with the *rrm3-CR, -6A* or *Δ86-110* mutants (Figs. 5D and EV4C,D), very nearly recapitulating the inviability of *dia2Δ rrm3Δ* cells (Fig. EV1D). This demonstrates that the Rrm3-Dpb2 interaction is critical for Rrm3 function in vivo, consistent with our analysis of Dpb2-binding mutants in in vitro DNA replication reactions (Figs. 5B and EV4B). Interestingly, *rrm3-2E* also exhibited synthetic sickness in combination with *dia2Δ* (Fig. 5C), albeit to a lesser extent than with the *rrm3Δ86-110* mutant. This is consistent with the observed CMGE binding defect of Rrm3-2E (Fig. 4D), and indicates that the Sld5-binding SLIM we have identified also partly contributes to Rrm3 function in vivo, despite being dispensable for replication termination in vitro (Fig. 5A). Taken together, these data also suggest that the Rrm3-CMGE interactions we have identified are critical for the Rrm3-driven pathway of CMG disassembly, which operates in *dia2Δ* cells (Polo Rivera et al, 2024).

## RTEL1 interacts with CMGE in a highly similar manner to Rrm3

The N-terminal IDR of Rrm3, which our data show mediates binding to CMGE, is not present in Pif1-family helicases in humans, mice, chickens, frogs, zebrafish or flies. Furthermore,

metazoa contain six other 5′-3′ helicases (RTEL1, FANCJ, DDX3, DDX11, HELB and XPD) in addition to PIF1, of which RTEL1 and HELB do not have clear homologues in yeast. Existing functional evidence from both *Xenopus* egg extracts and mammalian cells indicates that RTEL1 supports replisome progression past protein barriers and during replication termination (Campos et al, 2023; Sparks et al, 2019; Vannier et al, 2013), suggestive of a functional analogy with budding yeast Rrm3. RTEL1 binding to PCNA is required for normal replication fork progression in mouse embryonic fibroblasts, but the bypass of a leading strand DPC in *Xenopus* egg extracts appears to be independent of this interaction (Campos et al, 2023; Sparks et al, 2019; Vannier et al, 2013), suggesting additional mechanisms may exist for targeting RTEL1 to replication forks. Thus, the structural mechanisms for how any metazoan 5′-3′ helicases are recruited to replication forks remains incompletely understood in any species.

To assess if the mechanism of Rrm3 recruitment we have uncovered is conserved with any metazan 5′-3′ helicases, we used AlphaFold-Multimer to perform a targeted in silico screen to look for interactions between human 5′-3′ helicases and human POLE2 (the homologue of budding yeast Dpb2) and GINS (Appendix Table S2). Uniquely amongst these helicases, AlphaFold-Multimer predicts with high confidence that RTEL1 interacts with both POLE2 and GINS (Fig. 6A–C and Appendix Fig. S2A–D). The

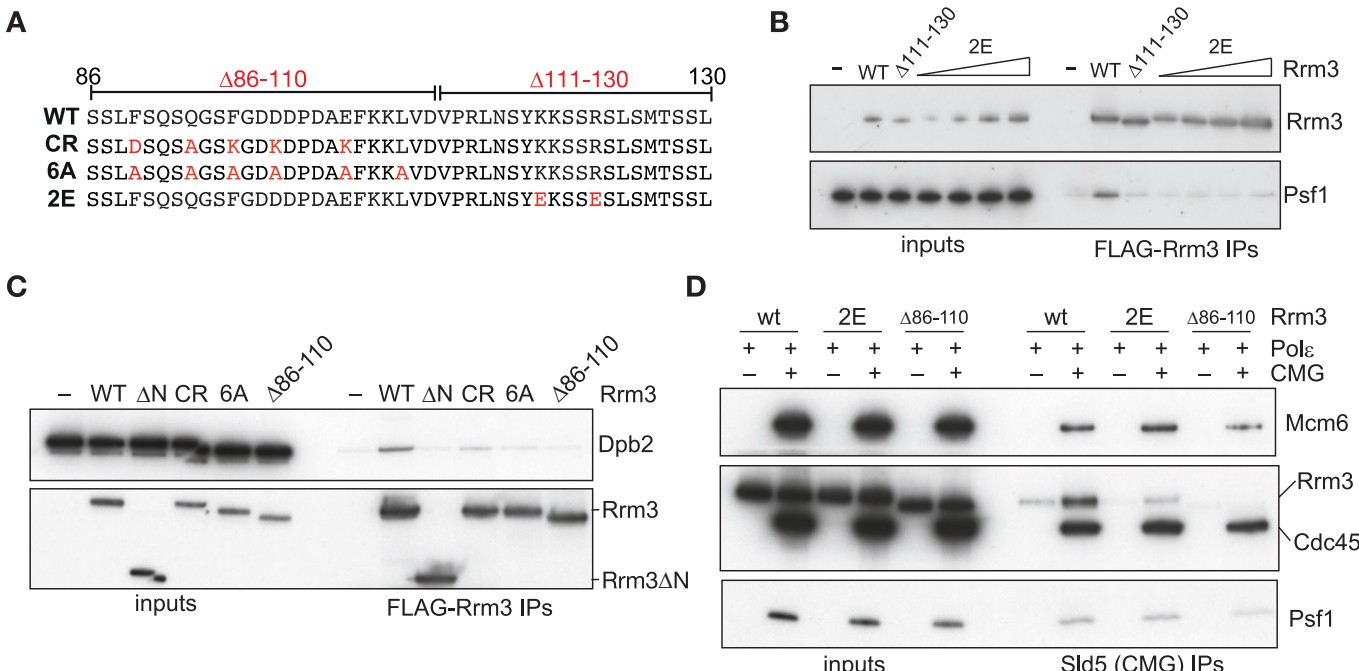

**Figure 4. Isolation of CMGE interaction mutants in Rrm3.**

(A) Schematic showing positions of mutations in Rrm3 residues 86–130, designed to disrupt binding to Polε (Rrm3Δ86-110, -CR and -6A) or GINS (Rrm3Δ111-130 and -2E). (B, C) Purified tetrameric GINS complex (B) or Polε (C) were mixed with FLAG-tagged wild-type Rrm3 or the indicated Rrm3 mutants. In (B), Rrm3-2E was included at 5, 10, 20 and 40 nM, whereas wild type and Rrm3Δ111-130 were included at 10 nM. Resultant complexes were isolated by anti-FLAG immunoprecipitation and detected by SDS-PAGE and immunoblotting. (D) The ability of wild type and the indicated mutant forms of Rrm3 to associate with CMG was monitored in the presence of Polε, as in Fig. 2A. Source data are available online for this figure.

predicted GINS and POLE2 interactions are mediated by three adjacent SLIMs in RTEL1, which are positioned in a largely unstructured region of the protein situated between the N-terminal helicase domain and the first of two harmonin-like domains (Fig. 6C). A short α-helix spanning RTEL1 residues 834–846 is predicted to bind POLE2, whilst the predicted interaction with GINS is mediated by two SLIMs that flank the POLE2-binding SLIM: residues 813–821 engage a short, slightly basic groove at the tip of PSF1 and SLD5, whilst residues 875–881 are predicted to form a short β-strand that interacts with SLD5 (Fig. 6A–C). These three SLIMs are well conserved in metazoa (Appendix Fig. S2E), and AlphaFold-Multimer predicts very similar interactions between RTEL1 and POLE2/GINS in mouse, chicken, frog and zebrafish (O. Olson and T. Deegan, unpublished observations).

The overall positioning of the predicted RTEL1 interacting sites on CMGE are strikingly similar to yeast Rrm3 (compare Figs. 3D, 6D). RTEL1 binds along a shallow cleft on the surface of POLE2 in a very similar position to Rrm3 in the budding yeast replisome (compare Fig. 3A,D with 6A,D), consistent with the critical role of this interaction in Rrm3 function (Fig. 5). In addition, the RTEL1-PSF1/SLD5 interaction site is directly adjacent to where Rrm3 binds Sld5, at the tip of the GINS tetramer (Figs. 3B,D and 6B,D). Furthermore, again like Rrm3, the spacing of the CMGE-interacting SLIMs in RTEL1 is complementary to their corresponding binding sites on CMGE (Fig. 6C,D), and would very likely enable RTEL1 to interact simultaneously with both POLE2 and two binding sites on GINS, to facilitate stable accessory helicase docking. As in the budding yeast replisome, the predicted RTEL1

binding sites on CMGE are well-positioned to facilitate access to the lagging strand DNA template (Fig. 6D), as it emerges from the replication fork junction between the MCM3 and MCM5 subunits of CMG.

To test the AlphaFold-Multimer predictions for RTEL1, we tested binding of purified human RTEL1 to POL ε and CMG individually, comparing wild-type RTEL1 with different RTEL1 mutants, in which the POLE2- and GINS-interacting SLIMs were deleted individually or in combination (Figs. 6 and EV5A). Wild-type FLAG-tagged RTEL1 interacted with POL ε, but this interaction was abolished by deletion of the POLE2-binding SLIM (RTELΔ834–846) (Fig. 7A). Likewise, wild-type FLAG-tagged RTEL1 interacted with CMG, but this interaction was slightly reduced with RTELΔ813–821, and completely abolished by deletion of RTEL1 residues 875–881, which are predicted to bind SLD5 (Fig. 7B). Deletion of the POLE2-binding SLIM did not affect the ability of RTEL1 to interact with CMG, and the RTEL1 GINS-binding mutants could still interact with POL ε (Fig. EV5B,C). Taken together, these data validate the AlphaFold-Multimer predictions presented in Fig. 6, and confirm that RTEL1 binds directly to CMG and POL ε via a very similar mechanism to Rrm3 in the budding yeast replisome.

## Discussion

Budding yeast Rrm3 and metazoan RTEL1 are specifically and transiently required to drive replisome progression when

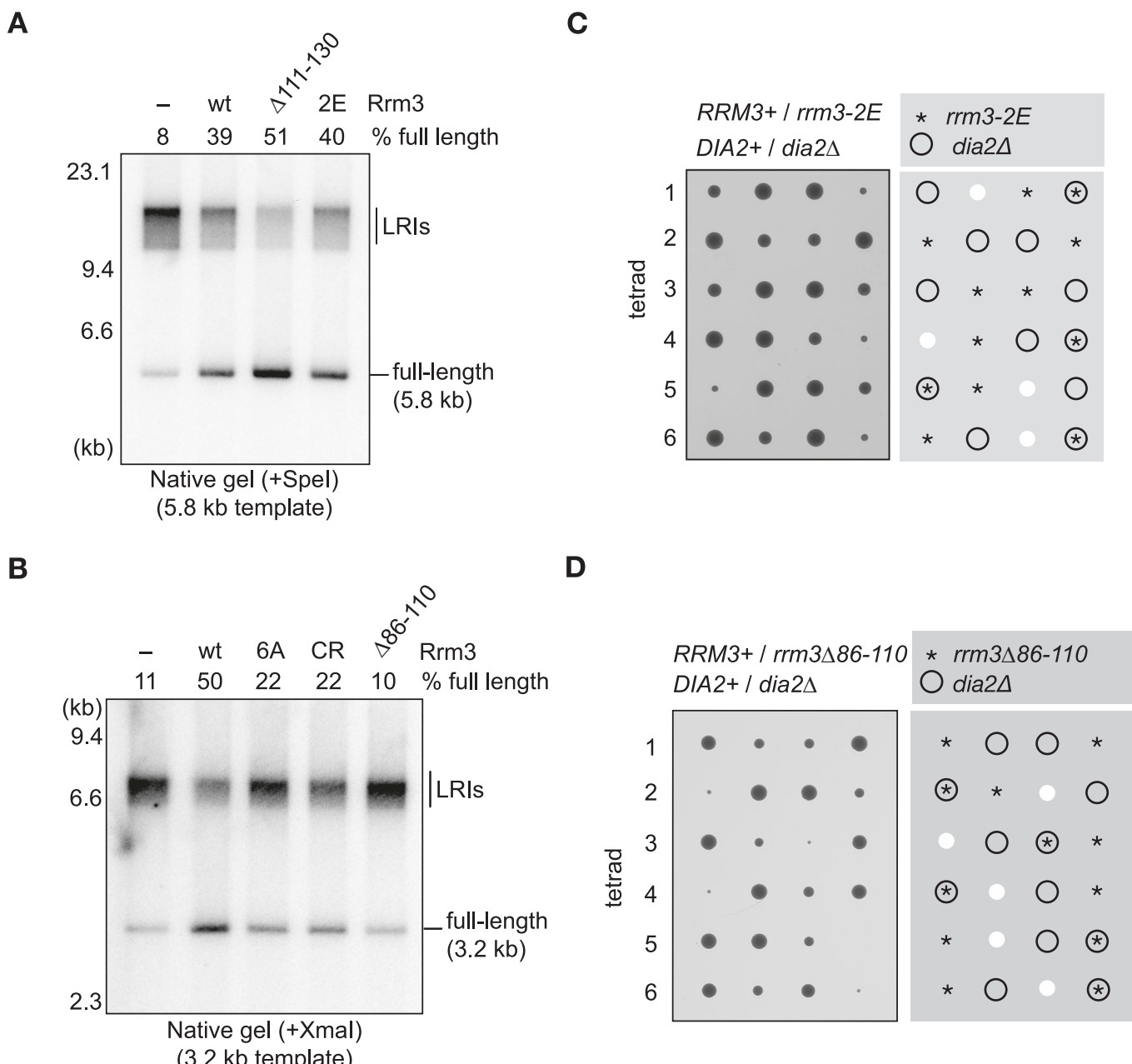

**Figure 5. CMGE binding is critical for Rrm3 function.**

(A, B) In vitro DNA replication reactions conducted as in Fig. 1B with wild-type Rrm3 or the indicated Rrm3 mutants. (C, D) Diploid yeast cells of the indicated genotypes were sporulated and the resulting tetrads were then dissected and grown on YPD medium for 2 days at 30 °C. Source data are available online for this figure.

unwinding by the CMG replicative helicase is blocked, for example at barriers on the leading strand template and during replication termination. In these instances, Rrm3/RTEL1 almost certainly translocate along the lagging strand DNA template with 5′-3′ polarity, thus moving with the same overall directionality as CMG (which translocates 3′-5′ on the leading strand template). Our discovery that both Rrm3 and RTEL1 interact with CMGE via Dpb2/POLE2 and GINS reveals a conserved mechanism for positioning accessory helicases proximally to the lagging strand template in the eukaryotic replisome. We also suggest that the

topological entrapment of the leading strand template by CMG (and the physical coupling of CMG to Polε) will limit the amount of accessible ssDNA on the leading strand template, thereby providing an additional level of strand specificity to Rrm3 and RTEL1.

Recent mass spectrometry analysis of replisome composition has indicated that Rrm3 is approximately as abundant as replisome components such as Tof1-Csm3, Ctf4 and Mrc1, when GINS (and therefore CMG) is immunoprecipitated from budding yeast cells in S-phase of the cell cycle (Reusswig et al, 2022). Furthermore,

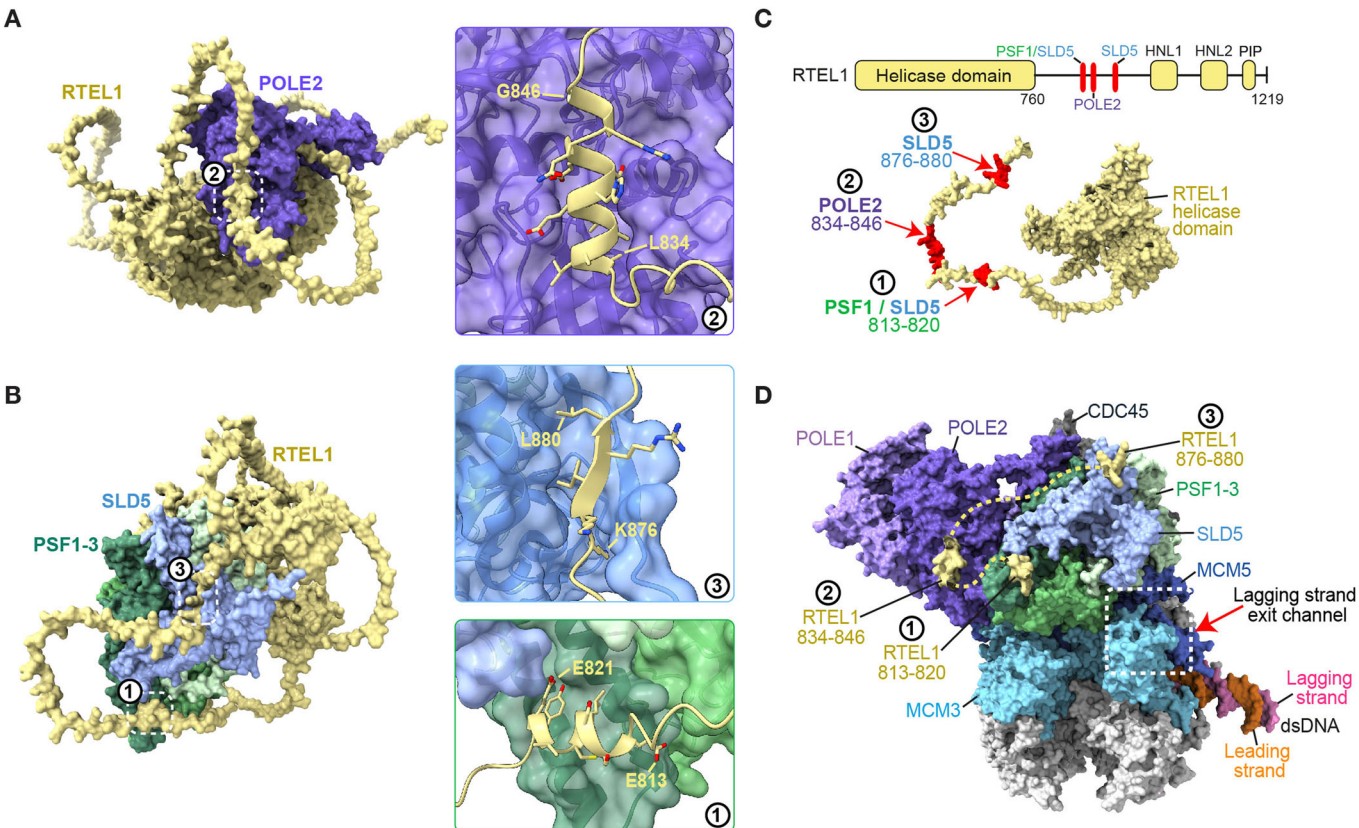

**Figure 6. Structural modelling of RTEL1 in the human replisome.**

(A, B) AlphaFold-Multimer models of *Homo sapiens* RTEL1 isoform 1 (NP_057518.1) bound to POLE2 (A) and the GINS tetramer (B). Interacting sites in RTEL1 are highlighted, and zoomed in views are shown to the right. Residue numbers relevant to RTEL1 deletion mutants (Fig. 7) are highlighted. (C) Domain structure and AlphaFold2-predicted monomer structure of RTEL1 isoform 1 (NP_057518.1). Positions of three adjacent POLE2- and GINS-binding Short Linear Interaction Motifs (SLIMs) in RTEL1 are indicated. Residues 890–1219, including the C-terminal harmonin-like (HNL1 and HNL2) domains (residues 890–975 and 1060–1140) and PCNA interaction (PIP) motif (residues 1166–1173), were removed from the AlphaFold2-predicted structure for simplicity. (D) POLE2- and GINS-binding SLIMs in RTEL1 (numbered as in (A–C)) were docked onto a cryo-EM structure of human CMG-Polε (PDB: 7PLO) by aligning on POLE2 and PSF1, respectively (see Methods for more details). Disordered segments of RTEL1 that connect the three SLIMs are represented as dashed lines. The path of the excluded lagging strand DNA template between MCM3 and MCM5 is indicated.

previous ChIP data have shown that Rrm3 associates with elongating replication forks (Azvolinsky et al, 2006). Taken together with the work presented herein, these data suggest that the CMGE-binding mechanism we have uncovered likely permits constitutive association of Rrm3 with the replisome (Fig. 7C, left). This association of an accessory helicase with CMGE during elongation would effectively 'prime' the replisome for efficient bypass of barriers, and help avoid prolonged fork stalling, which might result if an accessory helicase had to be recruited de novo every time CMG stalled.

It is possible that the Rrm3 helicase domain might occasionally bind to and translocate along the lagging strand template during normal elongation. However, such translocation is unlikely to have any significant impact on replisome progression, whilst CMG is actively engaged in unwinding at the replication fork junction. Alternatively, access of the Rrm3 helicase domain to the lagging strand template might be regulated by competition with other proteins, most notably Pol α-primase, which also bind to the lagging strand template during elongation. Relevant to this, recent cryo-EM structures of both yeast and human replisomes indicated

that Pol α-primase is recruited to the replisome via a series of interactions with Mcm3, Mcm5 and GINS, which position Pol α-primase at the leading edge of the replisome, directly adjacent to the lagging strand template (Jones et al, 2023). Thus, if Rrm3 does associate with elongating replisomes that have not stalled, access of its helicase domain to the lagging strand template might only occur when Pol α-primase disengages from DNA, as could occur if lagging strand priming momentarily stops upon CMG stalling (Fig. 7C, right). Further work, likely involving structural analysis of stalled replisomes, will be required to understand how accessory helicase unwinding on the lagging strand template is regulated during replisome progression and stalling.

Similarly to Rrm3, RTEL1 binding to CMGE could permit RTEL1 association with elongating replisomes, outside of CMG stalling. However, we note that the RTEL1-interacting surfaces we have identified on PSF1/SLD5 (Fig. 6, site 1) and SLD5 (Fig. 6, site 3) overlap with binding sites for POLA2 (Jones et al, 2023) and DONSON (Cvetkovic et al, 2023; Lim et al, 2023; Xia et al, 2023), respectively. To what extent competition between RTEL1 and Pol α-primase/DONSON for binding at these sites regulates RTEL1

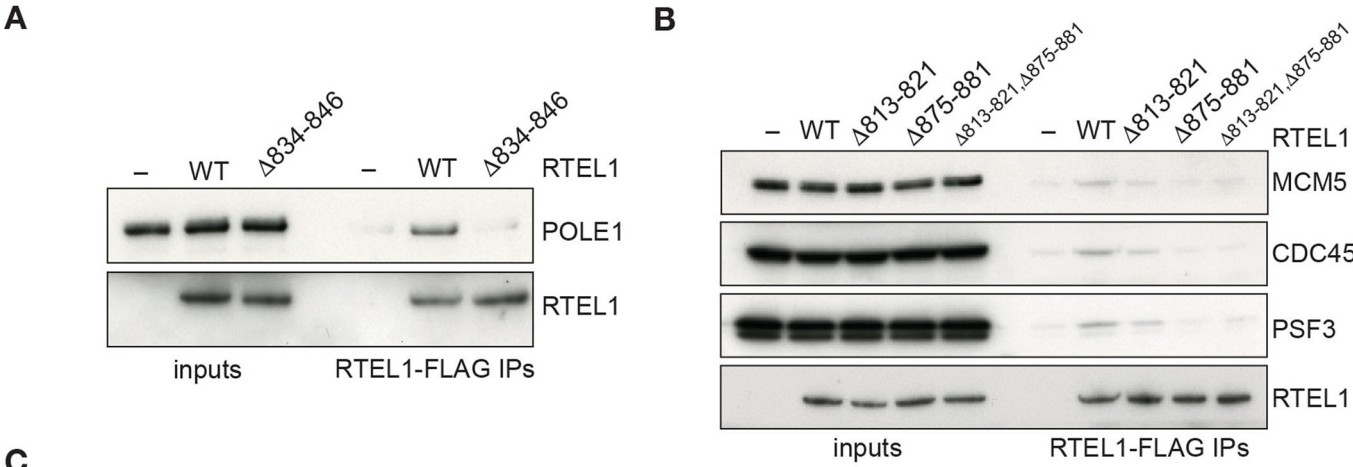

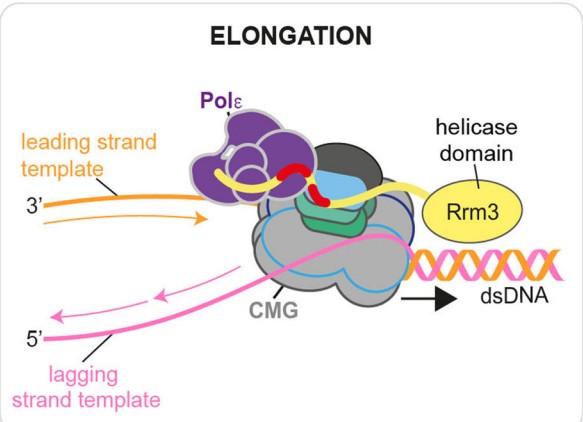

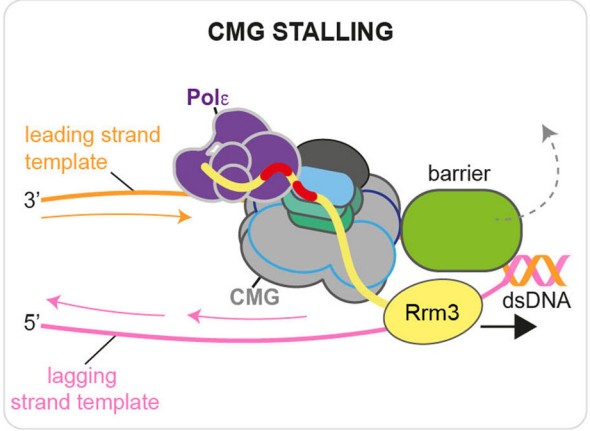

**Figure 7. Interaction of RTEL1 with POL ε and CMG.**

(A, B) Purified *Homo sapiens* POL ε (A) or CMG (B) were mixed with FLAG-tagged wild-type RTEL1 or the indicated RTEL1 mutants. Resultant complexes were isolated by anti-FLAG immunoprecipitation and detected by SDS-PAGE and immunoblotting. RTEL1 was detected by anti-FLAG immunoblotting. (C) Model for Rrm3 accessory helicase function during DNA replication. Dpb2- and Sld5-binding SLIMs in Rrm3 are shown in red. Removal of a protein barrier from the DNA (as an example of Rrm3 function) is depicted by a dashed line. We envisage that a similar mechanism could operate for RTEL1 in the human replisome, based on the CMGE binding mechanism we have identified. Discussed further in the text. Source data are available online for this figure.

recruitment to replication forks is an interesting avenue for future investigation. It will also be interesting to determine whether the previously identified RTEL1 interactions with PCNA, MCM10, TRF2, POLDIP3 and SLX4 (Bjorkman et al, 2020; Campos et al, 2023; Mendez-Bermudez et al, 2018; Takedachi et al, 2020; Vannier et al, 2013) and the RTEL1-CMGE interactions described in this work are part of the same or alternative mechanisms for RTEL1 recruitment to chromosomes. In any case, our data demonstrate that the previously suggested functional analogy between budding yeast Rrm3 and metazoan RTEL1 is mirrored by a conserved mode of CMGE interaction, via Dpb2/POLE2 and GINS. Notably, however, Rrm3 and RTEL1 do not exhibit any detectable homology in sequence or structure. Thus, the recruitment mechanism we have uncovered is indicative of convergent evolution, and reveals 5′-3′ accessory helicase docking onto CMGE as a previously unappreciated but conserved feature of eukaryotic replisome organisation.

## Methods

Resources and reagents from this study are detailed in the Reagents and Tools Table and will be made available on request. Requests should be made to Tom Deegan (tdeegan@ed.ac.uk).

### Yeast strains

The protein expression strains (Reagents and Tools Table) constructed in this study were generated by transforming the *Saccharomyces cerevisiae* strain yJF1 (MATa ade2-1 ura3-1 his3-11,15 trp1-1 leu2-3,112 can1-100 bar1Δ::hphNT pep4Δ::kanMX) with linearized plasmid (Reagents and Tools Table) using standard procedures. Genes for protein expression were codon optimized as previously described (Yeeles et al, 2015).

## Reagents and tools table

| Reagent/Resource | Reference or Source | Identifier or Catalog number |
|---|---|---|
| **Experimental models** | | |
| *Escherichia coli*: Rosetta™ (DE3) pLysS cells: F⁻ *ompT hsdS*ᴮ(rᴮ⁻ mᴮ⁻) *gal dcm* (DE3) pLysSRARE (Camᴿ) | Novagen | 70956 |
| *Escherichia coli* Stable Competent (High Efficiency) | New England Biolabs | C3040H |
| *Escherichia coli* 5-alpha Competent (High Efficiency) | New England Biolabs | C2987H |
| W303-1a: *MATa ade2-1 ura3-1 his3-11,15 trp1-1 leu2-3,112 can1-100* | Labib laboratory | W303-1a |
| yJF1: *MATa ade2-1 ura3-1 his3-11,15 trp1-1 leu2-3,112 can1-100 bar1::hphNT pep4::kanMx* | (Frigola et al, 2013) | N/A |
| ySDORC (ORC purification): *MATa ade2-1 ura3-1 his3-11,15 trp1-1 leu2-3,112 can1-100 bar1::hphNT pep4::kanMx his3::pRS303-ORC3 + ORC4 ura3::pRS306-CBP-TEV-ORC1 + ORC2 trp1::pRS304-ORC5 + ORC6* | (Frigola et al, 2013) | N/A |
| yAM33 (Cdt1-Mcm2-7 purification): *MATa ade2-1 ura3-1 his3-11,15 trp1-1 leu2-3,112 can1-100 bar1::hphNT pep4::kanMx his3::pRS303-CDT1 + GAL4 ura3::pRS306-MCM2 + CBP-TEV-MCM3 trp1::pRS304-MCM4 + MCM5 leu2::pRS305-MCM6 + MCM7* | (Coster et al, 2014) | N/A |
| ySDK8 (DDK purification): *MATa ade2-1 ura3-1 his3-11,15 trp1-1 leu2-3,112 can1-100 bar1::hphNT pep4::kanMX trp1::pRS304-CDC7 + CBP-TEV-DBF4* | (On et al, 2014) | N/A |
| yTD6 (Sld3-7 purification): *MATa ade2-1 ura3-1 his3-11,15 trp1-1 leu2-3,112 can1-100 bar1::hphNT pep4::kanMX leu2::pRS305-SLD7 his3::pRS303-SLD3-TCP + GAL4* | (Yeeles et al, 2015) | N/A |
| yTD8 (Sld2 purification): *MATa ade2-1 ura3-1 his3-11,15 trp1-1 leu2-3,112 can1-100 bar1::hphNT pep4::kanMX his3::pRS303-SLD2-3FLAG(nat-NT2) + GAL4* | (Yeeles et al, 2015) | N/A |
| yJY13 (Cdc45 purification): *MATa ade2-1 ura3-1 his3-11,15 trp1-1 leu2-3,112 can1-100 bar1::hphNT pep4::kanMX his3::pRS303-CDC45-iFLAG2 + GAL4* | (Yeeles et al, 2015) | N/A |
| yJY26 (Dpb11 purification): *MATa ade2-1 ura3-1 his3-11,15 trp1-1 leu2-3,112 can1-100 bar1::hphNT pep4::kanMX his3::pRS303-DPB11-3FLAG(nat-NT2) + GAL4* | (Yeeles et al, 2015) | N/A |
| yAJ2 (Pol ε purification): *MATa ade2-1 ura3-1 his3-11,15 trp1-1 leu2-3,112 can1-100 bar1::hphNT pep4::kanMX trp1::pRS304-POL2 + DPB4-TEV-CBP ura3::pRS306DPB2 + DPB3* | (Yeeles et al, 2015) | N/A |
| yAE37 (S-CDK purification): *MATa ade2-1 ura3-1 his3-11,15 trp1-1 leu2-3,112 can1-100 bar1::hphNT pep4::kanMX ura3::pRS306-CKS1 + CDC28 his3::pRS303-CBP-TEV-CLB5 + GAL4* | (Yeeles et al, 2015) | N/A |
| yAE40 (Ctf4 purification): *MATa ade2-1 ura3-1 his3-11,15 trp1-1 leu2-3,112 can1-100 bar1::hphNT pep4::kanMX his3::pRS303-CBP-TEV-CTF4 + GAL4* | (Yeeles et al, 2015) | N/A |
| yJY23 (Pol α/primase purification): *MATa ade2-1 ura3-1 his3-11,15 trp1-1 leu2-3,112 can1-100 bar1::hphNT pep4::kanMX trp1::pRS304-POL1 + POL12 ura3:: pRS306-CBP-TEV-PRI1 + PRI2* | (Yeeles et al, 2015) | N/A |
| yAE34 (Pol δ purification): *MATa ade2-1 ura3-1 his3-11,15 trp1-1 leu2-3,112 can1-100 bar1::hphNT pep4::kanMX ura3::pRS306-POL31 + POL3 his3::pRS303-POL32-CBP + GAL4* | (Yeeles et al, 2015) | N/A |
| yAE41 (RFC purification): *MATa ade2-1 ura3-1 his3-11,15 trp1-1 leu2-3,112 can1-100 bar1::hphNT pep4::kanMX ura3::pRS306-RFC2 + CBP-RFC3 trp1::pRS304-RFC4 + RFC5 his3::pRS303-RFC1 + GAL4* | (Yeeles et al, 2015) | N/A |
| yAE71 (Mrc1 purification): *MATa ade2-1 ura3-1 his3-11,15 trp1-1 leu2-3,112 can1-100 bar1::hphNT pep4::kanMX his3::pRS303-MRC1-5FLAG* | (Deegan et al, 2019) | N/A |
| yAE31 (RPA purification): *MATa ade2-1 ura3-1 his3-11,15 trp1-1 leu2-3,112 can1-100 bar1::hphNT pep4::kanMX his3::pRS303-CBP-TEV-RFA1 + GAL4 ura3::pRS306-RFA2 + RFA3* | (Yeeles et al, 2015) | N/A |
| yTDK4 (Tof1-Csm3 purification): *MATa ade2-1 ura3-1 his3-11,15 trp1-1 leu2-3,112 can1-100 bar1Δ::hphNT pep4Δ::kanMX leu2::pRS305-TOF1 + CBP-TEV-CSM3* | (Deegan et al, 2019) | N/A |
| yTDK6 (Top1 purification): *MATa ade2-1 ura3-1 his3-11,15 trp1-1 leu2-3,112 can1-100 bar1Δ::hphNT pep4Δ::kanMX leu2::pRS305-CBP-TEV-TOP1* | (Deegan et al, 2019) | N/A |
| yJY31 (Fen1 purification): *MATa ade2-1 ura3-1 his3-11,15 trp1-1 leu2-3,112 can1-100 bar1Δ::hphNT pep4Δ::kanMX leu2::pRS305-FEN1-2FLAG* | (Guillam et al, 2020) | N/A |
| yJY33 (Cdc9 purification): *MATa ade2-1 ura3-1 his3-11,15 trp1-1 leu2-3,112 can1-100 bar1Δ::hphNT pep4Δ::kanMX leu2::pRS305-CDC9-2FLAG* | (Guillam et al, 2020) | N/A |

| Reagent/Resource | Reference or Source | Identifier or Catalog number |
|---|---|---|
| yTDK20 (CMG purification): *MATa/MATα pep4Δ::kanMX/pep4Δ::kanMX bar1Δ::hph-NT1/bar1Δ::hph-NT1 ade2-1/ade2-1 ura3-1/ura3-1::pRS306-MCM2-GAL1,10-CBP-TEV-MCM3 his3-11::pRS303-CDC45iFLAG2-GAL1,10-GAL4/his3-11 leu2-3::pRS305-PSF2-GAL1,10-PSF3/leu2-3::pRS305-MCM7-GAL1,10-MCM6 trp1-1::pRS304-PSF1-GAL1,10-SLD5/trp1-1::pRS304-MCM5-GAL1,10-MCM4 ctf4-I901E/ctf4-I901E* | (Deegan et al, 2020) | N/A |
| yTDK9 (Rrm3 purification): *MATa ade2-1 ura3-1 his3-11,15 trp1-1 leu2-3,112 can1-100 bar1Δ::hphNT pep4Δ::kanMX leu2::pRS305-3FLAG-TEV-RRM3* | (Deegan et al, 2019) | N/A |
| yTDK11 (Rrm3ΔN purification): *MATa ade2-1 ura3-1 his3-11,15 trp1-1 leu2-3,112 can1-100 bar1Δ::hphNT pep4Δ::kanMX leu2::pRS305-2FLAG-RRM3ΔN* | Labib laboratory | N/A |
| yOO4 (Rrm3-2E purification): *MATa ade2-1 ura3-1 his3-11,15 trp1-1 leu2-3,112 can1-100 bar1::hphNT pep4::kanMx leu2::pRS305-3FLAG-TEV-RRM3-2E* | This study | N/A |
| yOO5 (Rrm3Δ111-130 purification): *MATa ade2-1 ura3-1 his3-11,15 trp1-1 leu2-3,112 can1-100 bar1::hphNT pep4::kanMx leu2::pRS305-3FLAG-TEV-RRM3-Δ111-130* | This study | N/A |
| yOO7 (Rrm3Δ86-110 purification): *MATa ade2-1 ura3-1 his3-11,15 trp1-1 leu2-3,112 can1-100 bar1::hphNT pep4::kanMx leu2::pRS305-3FLAG-TEV RRM3-Δ86-110* | This study | N/A |
| yOO8 (Rrm3-6A purification): *MATa ade2-1 ura3-1 his3-11,15 trp1-1 leu2-3,112 can1-100 bar1::hphNT pep4::kanMx leu2::pRS305-3FLAG-TEV-RRM3-6A* | This study | N/A |
| yOO9 (Rrm3-CR purification): *MATa ade2-1 ura3-1 his3-11,15 trp1-1 leu2-3,112 can1-100 bar1::hphNT pep4::kanMx leu2::pRS305-3FLAG-TEV-RRM3-CR* | This study | N/A |
| yOO11 (Rrm3N-BacPif1 purification): *MATa ade2-1 ura3-1 his3-11,15 trp1-1 leu2-3,112 can1-100 bar1::hphNT pep4::kanMx leu2::pRS305-3FLAG-TEV-RRM3N-BACPIF1* | This study | N/A |
| yOO12 (BacPif1 purification): *MATa ade2-1 ura3-1 his3-11,15 trp1-1 leu2-3,112 can1-100 bar1::hphNT pep4::kanMx leu2::pRS305-3FLAG-TEV-BACPIF1* | This study | N/A |
| yKL2713 (dia2Δ/DIA2+ diploid): *MATa ade2-1 ura3-1 his3-11,15 trp1-1 leu2-3,112 can1-100 / MATα ade2-1 ura3-1 his3-11,15 trp1-1 leu2-3,112 can1-100 dia2Δ::HIS3/DIA2* | Labib laboratory | N/A |
| yKL14026 (mec1Δ sml1Δ haploid): MATa ade2-1 ura3-1 his3-11,15 trp1-1 leu2-3,112 can1-100 mec1Δ::ADE2 sml1Δ::kanMX | Labib laboratory | N/A |
| yHM28 (dia2Δ haploid): *MATa ade2-1 ura3-1 his3-11,15 trp1-1 leu2-3,112 can1-100 dia2Δ::HIS3* | (Morohashi et al, 2009) | N/A |
| yOO24 (dia2Δ x rrm3Δ86-110 diploid): *MATa ade2-1 ura3-1 his3-11,15 trp1-1 leu2-3,112 can1-100 / MATα ade2-1 ura3-1 his3-11,15 trp1-1 leu2-3,112 can1-100 dia2Δ::HIS3/DIA2 rrm3Δ86-110::URA3/RRM3* | This study | N/A |
| yOO28 (dia2Δ x rrm3-6A diploid): *MATa ade2-1 ura3-1 his3-11,15 trp1-1 leu2-3,112 can1-100 / MATα ade2-1 ura3-1 his3-11,15 trp1-1 leu2-3,112 can1-100 dia2Δ::HIS3/DIA2 rrm3-6A::URA3/RRM3* | This study | N/A |
| yOO29 (dia2Δ x rrm3-CR diploid): *MATa ade2-1 ura3-1 his3-11,15 trp1-1 leu2-3,112 can1-100 / MATα ade2-1 ura3-1 his3-11,15 trp1-1 leu2-3,112 can1-100 dia2Δ::HIS3/DIA2 rrm3-CR::URA3/RRM3* | This study | N/A |
| yOO35 (dia2Δ x rrm3-2E diploid): *MATa ade2-1 ura3-1 his3-11,15 trp1-1 leu2-3,112 can1-100 / MATα ade2-1 ura3-1 his3-11,15 trp1-1 leu2-3,112 can1-100 dia2Δ::HIS3/DIA2 rrm3-2E::URA3/RRM3* | This study | N/A |
| yTDE9 (mec1Δ sml1Δ rrm3ΔN diploid): MATa ade2-1 ura3-1 his3-11,15 trp1-1 leu2-3,112 can1-100 / MATα ade2-1 ura3-1 his3-11,15 trp1-1 leu2-3,112 can1-100 mec1Δ::ADE2/MEC1 sml1Δ::kanMX/SML1 RRM3/rrm3ΔN::URA3 | This study | N/A |
| yTDE10 (mec1Δ sml1Δ rrm3Δ diploid): MATa ade2-1 ura3-1 his3-11,15 trp1-1 leu2-3,112 can1-100 / MATα ade2-1 ura3-1 his3-11,15 trp1-1 leu2-3,112 can1-100 mec1Δ::ADE2/MEC1 sml1Δ::kanMX/SML1 RRM3/rrm3Δ::hphNT | This study | N/A |
| yTDE11 (sml1Δ rrm3ΔN haploid): MATα ade2-1 ura3-1 his3-11,15 trp1-1 leu2-3,112 can1-100 rrm3ΔN::URA3 sml1Δ::kanMX | This study | N/A |
| yTDE12 (sml1Δ rrm3Δ haploid): MATα ade2-1 ura3-1 his3-11,15 trp1-1 leu2-3,112 can1-100 rrm3Δ::hphNT sml1Δ::kanMX | This study | N/A |
| yTDE13 (mec1Δ sml1Δ/sml1Δ rrm3Δ diploid): MATa ade2-1 ura3-1 his3-11,15 trp1-1 leu2-3,112 can1-100 sml1Δ::kanMX / MATα ade2-1 ura3-1 his3-11,15 trp1-1 leu2-3,112 can1-100 mec1Δ::ADE2/MEC1 sml1Δ::kanMX/sml1Δ::kanMX RRM3/rrm3Δ::hphNT | This study | N/A |
| yTDE14 (mec1Δ sml1Δ/sml1Δ rrm3ΔN diploid): MATa ade2-1 ura3-1 his3-11,15 trp1-1 leu2-3,112 can1-100 / MATα ade2-1 ura3-1 his3-11,15 trp1-1 leu2-3,112 can1-100 mec1Δ::ADE2/MEC1 sml1Δ::kanMX/sml1Δ::kanMX RRM3/rrm3ΔN::URA3 | This study | N/A |
| yHM128 (rrm3Δ x dia2Δ diploid): *MATa ade2-1 ura3-1 his3-11,15 trp1-1 leu2-3,112 can1-100 / MATα ade2-1 ura3-1 his3-11,15 trp1-1 leu2-3,112 can1-100 dia2Δ::HIS3/DIA2 rrm3Δ::hphNT/RRM3* | (Morohashi et al, 2009) | N/A |
| yKL1: *MATa ade2-1 ura3-1 his3-11,15 trp1-1 leu2-3,112 can1-100 / MATα ade2-1 ura3-1 his3-11,15 trp1-1 leu2-3,112 can1-100* | Labib laboratory | N/A |

| Reagent/Resource | Reference or Source | Identifier or Catalog number |
|---|---|---|
| yEH9 (rrm3ΔN diploid): *MATa ade2-1 ura3-1 his3-11,15 trp1-1 leu2-3,112 can1-100 / MATα ade2-1 ura3-1 his3-11,15 trp1-1 leu2-3,112 can1-100 RRM3 / rrm3ΔN::URA3* | This study | N/A |
| yEH10 (rrm3ΔN haploid): *MATα ade2-1 ura3-1 his3-11,15 trp1-1 leu2-3,112 can1-100 rrm3ΔN::URA3* | This study | N/A |
| yEH29 (rrm3Δ haploid): *MATα ade2-1 ura3-1 his3-11,15 trp1-1 leu2-3,112 can1-100 rrm3Δ::hphNT* | This study | N/A |
| yOO1 (RTEL1 purification): *MATa ade2-1 ura3-1 his3-11,15 trp1-1 leu2-3,112 can1-100 bar1::hphNT pep4::kanMx leu2::pRS305-RTEL1-2FLAG* | This study | N/A |
| ySP075 (RTEL1ΔE813-E821 purification): *MATa ade2-1 ura3-1 his3-11,15 trp1-1 leu2-3,112 can1-100 bar1::hphNT pep4::kanMx leu2-3 :: LEU2pRS305-RTEL1ΔE813-E821_2XFLAG* | This study | N/A |
| ySP072 (RTEL1ΔL834-G846 purification): *MATa ade2-1 ura3-1 his3-11,15 trp1-1 leu2-3,112 can1-100 bar1::hphNT pep4::kanMx leu2-3 :: LEU2pRS305-RTEL1ΔL834-G846_2XFLAG* | This study | N/A |
| ySP076 (RTEL1ΔR875-V881 purification): *MATa ade2-1 ura3-1 his3-11,15 trp1-1 leu2-3,112 can1-100 bar1::hphNT pep4::kanMx leu2-3 :: LEU2pRS305-RTEL1ΔR875-V881_2XFLAG* | This study | N/A |
| ySP070 (RTEL1 ΔE813-E821 ΔR875-V881 purification): *MATa ade2-1 ura3-1 his3-11,15 trp1-1 leu2-3,112 can1-100 bar1::hphNT pep4::kanMx leu2-3 :: LEU2pRS305-RTEL1 ΔE813-E821 ΔR875-V881_2XFLAG* | This study | N/A |
| ySP041 (*hs*CMG purification): *MATa / MATα leu2-3 :: LEU2pRS305-HsMcm4-HsMcm5 / leu2-3 :: LEU2pRS305-HsPsf2-HsPsf3 ura3-1 :: URA3pRS306-HsMcm2-HsMcm3 / ura3-1 :: URA3pRS306-PrA-3TEV-HsSld5-HsPsf1 his3-11::HIS3pRS303-Mcmbp_cbp / his3-11 :: HIS3pRS303-HsCdc45 trp1-1 :: TRP1pRS304-HsMcm6-HsMcm7 / trp1-1 bar1Δ :: HphNT / bar1Δ :: HphNT pep4Δ :: kanMX / pep4Δ :: kanMX* | This study | N/A |
| ySP061 (*hs*POLε purification): *MATa ade2-1 ura3-1 his3-11,15 trp1-1 leu2-3,112 can1-100 pep4Δ::ADE2 ura3-1 :: URA3pRS306-PolE1_PolE2_tev_2XFLAG trp1-1 :: TRP1pRS304-PolE3_PolE4* | This study | N/A |
| **Recombinant DNA** | | |
| pTDK15 (generation of pOO8, pOO10 and pOO12-16) | (Deegan et al, 2019) | N/A |
| pOO8 (Gal1,10-3XFLAG-TEV-Rrm3<sup>K118E, R122E</sup> for yOO4 generation and Rrm3-2E purification. Cloned by Gibson assembly into pTDK15) | This study | N/A |
| pOO10 (Gal1,10-3XFLAG-TEV-Rrm3-Δ111-130 for yOO5 generation and Rrm3Δ111-130 purification. Cloned by Gibson assembly into pTDK15) | This study | N/A |
| pOO12 (Gal1,10-3XFLAG-TEV-Rrm3N-BacPif1 for yOO11 generation and Rrm3N-BacPif1 purification. Cloned by Gibson assembly into pTDK15) | This study | N/A |
| pOO13 (Gal1,10-3XFLAG-TEV-BacPif1 for yOO12 generation and BacPif1 purification. Cloned by Gibson assembly into pTDK15) | This study | N/A |
| pOO14 (Gal1,10-3XFLAG-TEV-Rrm3-Δ86-110 for yOO7 generation and Rrm3Δ86-110 purification. Cloned by Gibson assembly into pTDK15) | This study | N/A |
| pOO15 (Gal1,10-3XFLAG-TEV-Rrm3<sup>F89D, Q93A, F96K, D99K, E104K</sup> for yOO9 generation and Rrm3-CR purification. Cloned by Gibson assembly into pTDK15) | This study | N/A |
| pOO16 (Gal1,10-3XFLAG-TEV-Rrm3<sup>F89A, Q93A, F96A, D99A, E104A, L108A</sup> for yOO8 generation and Rrm3-6A purification. Cloned by Gibson assembly into pTDK15) | This study | N/A |
| pTDK48 (for pTDE8 generation) | (Jenkyn-Bedford et al, 2021) | N/A |
| pTDE8 (WT Rrm3 amplified from yeast gDNA using primers TD281 and TD282 cloned into pTDK48 at XmaI (5') and AscI (3') restriction sites (replacing Dia2-13A) for pTDE9-12 generation and yEH9 and yEH10 strain construction) | This study | N/A |
| pTDE9 (Rrm3-2E mutations introduced into pTDE8 by Gibson assembly for pOO24 generation) | This study | N/A |
| pTDE10 (Rrm3-6A mutations introduced into pTDE8 by Gibson assembly for pOO25 generation) | This study | N/A |
| pTDE11 (Rrm3-CR mutations introduced into pTDE8 by Gibson assembly for pOO26 generation) | This study | N/A |
| pTDE12 (Rrm3Δ86-110 mutations introduced into pTDE8 by Gibson assembly for pOO27 generation) | This study | N/A |
| pOO24 (1 kb yeast genomic DNA 5' Rrm3 amplified using primers TD379 and TD380 and cloned into pTDE9 at BsiWI (5') and XmaI (3') restriction sites) | This study | N/A |
| pOO25 (1 kb yeast genomic DNA 5' Rrm3 amplified using primers TD379 and TD380 and cloned into pTDE10 at BsiWI (5') and XmaI (3') restriction sites for Rrm3-6A (yOO28) strain construction) | This study | N/A |
| pOO26 (1 kb yeast genomic DNA 5' Rrm3 amplified using primers TD379 and TD380 and cloned into pTDE11 at BsiWI (5') and XmaI (3') restriction sites for Rrm3-CR (yOO29) strain construction) | This study | N/A |

| Reagent/Resource | Reference or Source | Identifier or Catalog number |
|---|---|---|
| pOO27 (1 kb yeast genomic DNA 5′ Rrm3 amplified using primers TD379 and TD380 and cloned into pTDE12 at BsiWI (5′) and XmaI (3′) restriction sites for Rrm3Δ86-110 (yOO24) strain construction) | This study | N/A |
| pOO32 (Site-directed mutagenesis of pOO24 for Rrm3-2E (yOO35) strain construction) | This study | N/A |
| pTD195 (Gal1,10-RTEL1_2XFLAG. Cloned by Gibson assembly into pTDK15) | This study | N/A |
| pSP51 (Gal1,10-RTEL1ΔE813-E821_2XFLAG. Cloned by Gibson assembly into pTD195) | This study | N/A |
| pSP48 (Gal1,10-RTEL1ΔL834-G846_2XFLAG. Cloned by Gibson assembly into pTD195) | This study | N/A |
| pSP52 (Gal1,10-RTEL1ΔR875-V881_2XFLAG. Cloned by Gibson assembly into pTD195) | This study | N/A |
| pSP53 (Gal1,10-RTEL1ΔE813-E821 ΔR875-V881_2XFLAG. Cloned by Gibson assembly into pTD195) | This study | N/A |
| pSP29 (hsPolE1-Gal1,10-hsPOLE2_TEV_2XFLAG. Cloned by Gibson assembly) | This study | N/A |
| pSP30 (hsPolE3-Gal1,10-hsPOLE4. Cloned by Gibson assembly) | This study | N/A |
| pGEX-6p1 Cdc6 (Cdc6 purification) | (Frigola et al, 2013) | N/A |
| pET28a PCNA (PCNA purification) | (Yeeles et al, 2015) | N/A |
| pJFDJ5 (GINS purification) | (Yeeles et al, 2015) | N/A |
| pJL005 (GINS with TwinStrep-Psf3 purification) | (Lewis et al, 2022) | N/A |
| pBP6 HIS6-Mcm10 (Mcm10 purification) | (Yeeles et al, 2015) | N/A |
| pBS/ARS1 WTA (3.2 kb replication template) | (Marahrens and Stillman, 1992) | N/A |
| pCFK1_WT (5.8 kb replication template) | (Yeeles et al, 2015) | N/A |
| λ DNA-HindIII Digest: molecular weight markers | New England Biolabs | N3012S |
| **Antibodies** | | |
| Anti-FLAG (M2) | Sigma-Aldrich | F3615 |
| Anti-Psf1 (S. cerevisiae) | MRC PPU Reagents and Services | DU73858 |
| Anti-Sld5 (S. cerevisiae) | MRC PPU Reagents and Services | DU73863 |
| Anti-Mcm6 (S. cerevisiae) | MRC PPU Reagents and Services | DU62612 |
| Anti-Pol2 (S. cerevisiae) | (Deegan et al, 2020) | N/A |
| Anti-Dpb2 (S. cerevisiae) | MRC PPU Reagents and Services | DU47651 |
| Anti-sheep HRP | Sigma | A3415 |
| Anti-mouse HRP | Vector labs | PI-2000 |
| Anti-FLAG | Sigma-Aldrich | F7425 |
| Anti-MCM5 (M. musculus) | MRC PPU Reagents and Services | DU51792 |
| Anti-CDC45 (M. musculus) | MRC PPU Reagents and Services | DU35753 |

| Reagent/Resource | Reference or Source | Identifier or Catalog number |
|---|---|---|
| Anti-PSF3 (*M. musculus*) | MRC PPU Reagents and Services | DU24601 |
| Anti-POLE1 (*M. musculus*) | MRC PPU Reagents and Services | DU27959 |
| **Oligonucleotides and other sequence-based reagents** | | |
| Fwd primer for amplification of pTDK15 backbone for construction of pOO8 and pOO10 by Gibson assembly: TD150: GCCATTGTTGAGAAAGACCG | Integrated DNA Technologies (IDT) | N/A |
| Rev primer for amplification of pTDK15 backbone for construction of pOO8 and pOO10 by Gibson assembly: TD151: GGTAGAGTTCTTGGAAGCAGC | Integrated DNA Technologies (IDT) | N/A |
| Fwd primer for amplification of pTDK15 backbone for construction of pOO14-16 by Gibson assembly: TD358: GGTTTGAAGTTGACTGTTCC | Integrated DNA Technologies (IDT) | N/A |
| Rev primer for amplification of pTDK15 backbone for construction of pOO14-16 by Gibson assembly: TD359: TGGAACCGTTAGAGGATCTC | Integrated DNA Technologies (IDT) | N/A |
| Fwd primer for amplification of pTDK15 backbone for construction of pOO12 and pOO13 by Gibson assembly: TD234: TAAATTGAATTGAATTGAAATCG | Integrated DNA Technologies (IDT) | N/A |
| Rev primer for amplification of pTDK15 backbone for construction of pOO12 by Gibson assembly: TD235: TCAAGACGACTGGGGAAG | Integrated DNA Technologies (IDT) | N/A |
| Rev primer for amplification of pTDK15 backbone for construction of pOO13 by Gibson assembly: TD236: TCCTTGTCATCATCGTCC | Integrated DNA Technologies (IDT) | N/A |
| Fwd primer for amplification of WT Rrm3 from yeast gDNA: TD281: TGTAACCCGGGATGTTCAGGTCGCATGCC | Integrated DNA Technologies (IDT) | N/A |
| Rev primer for amplification of WT Rrm3 from yeast gDNA: TD282: GATCTGGCGCGCCTCATTTCAAAGTTTCTAAACGTTTATAG | Integrated DNA Technologies (IDT) | N/A |
| Fwd primer for amplification of Rrm3-URA3 cassette from pTDE8 for endogenous gene replacement: TD283: GAGGAGAACAAGCTCAAAAGTCGAGAGATTTGTTCTTATAAGACATCCCGATGTTCAGGTCGCATGCCTC | Integrated DNA Technologies (IDT) | N/A |
| Fwd primer for amplification of Rrm3-URA3 cassette from pOO24-27 for endogenous gene replacement: TD371: CGTACGCATAGAACCGAGTGTAACACC | Integrated DNA Technologies (IDT) | N/A |
| Rev primer for amplification of Rrm3-URA3 cassette from pTDE8 and pOO24-27 for endogenous gene replacement: TD284: AACAAGAAAAGAAAACTTCAACTAGAGTATATGCATT TATTCGTTGCAAGATCGATGAATTCGAGCTCGATTA | Integrated DNA Technologies (IDT) | N/A |
| Fwd primer for amplification of Rrm3ΔN-URA3 cassette from pTDE8 for endogenous gene replacement: TD301: GAGGAGAACAAGCTCAAAAGTCGAGAGATTTGTTCTTATAAGACATCCCGATGGAGTTTCAAGGTTTAAAGC | Integrated DNA Technologies (IDT) | N/A |
| Fwd primer for checking integration of Rrm3-URA3 cassette at Rrm3 locus and sequencing: TD306: CGTTGGTGGTATGACTAAATTG | Integrated DNA Technologies (IDT) | N/A |
| Rev primer for checking integration of Rrm3-URA3 cassette at Rrm3 locus: TD307: GTGTAGGATCTGATTTCCCTCAC | Integrated DNA Technologies (IDT) | N/A |
| Fwd primer for checking integration of Rrm3-URA3 cassette at Rrm3 locus and sequencing: TD314: CCTCAGTGGCAAATCCTAACC | Integrated DNA Technologies (IDT) | N/A |
| Rev primer for amplification of pTDE8 backbone for construction of pTDE9-12 by Gibson assembly: TD320: AGTCCATAAGCTTATCGTCG | Integrated DNA Technologies (IDT) | N/A |

| Reagent/Resource | Reference or Source | Identifier or Catalog number |
|---|---|---|
| Fwd primer for amplification of pTDE8 backbone for construction of pTDE9-12 by Gibson assembly: TD321: GGTTTAAAGCTCACAGTACC | Integrated DNA Technologies (IDT) | N/A |
| Fwd primer for amplification of 1 kb yeast gDNA 5′ of Rrm3 for insertion into pTDE9-12: TD379: ATATATCGTACGCATAGAACCGAGTGTAACACC | Integrated DNA Technologies (IDT) | N/A |
| Fwd primer for site-directed mutagenesis of pTDE9 to generate pOO32: TD409: GAACATCGGGATGTCTTATAAGAAC | Integrated DNA Technologies (IDT) | N/A |
| Fwd primer for site-directed mutagenesis of pTDE9 to generate pOO32: TD410: TCCCGATGTTCAGGTCACATG | Integrated DNA Technologies (IDT) | N/A |
| Rev primer for amplification of 1 kb yeast gDNA 5′ of Rrm3 for insertion into pTDE9-12: TD380: ATATATCCCGGGATGTCTTATAAGAAC | Integrated DNA Technologies (IDT) | N/A |
| Leading strand oligo for helicase assay substrate (25 bp duplex): TD254: 5′-GTGATTAGAGAATTGGAGAGTGTGTTTTTTTTTTTTT TTTTTTTTTTTTTTTTTTTTTT*T*T*T*T*T-3′ | Integrated DNA Technologies (IDT) | N/A |
| Lagging strand oligo for helicase assay substrate (25 bp duplex): TD255: 5′-GACAAGAAGGGAACAGACAGCGACACACTCTCCAATTCTCTAATCAC-3′ | Integrated DNA Technologies (IDT) | N/A |
| Amplify RTEL1 backbone to insert mutants gblocks, TD390 5′-TGCAAATCCAAAGACTTAGCC-3′ | Integrated DNA Technologies (IDT) | N/A |
| Amplify RTEL1 backbone to insert mutants gblocks, TD406 5′-CCAGAAGAACCAGTCGC-3′ | Integrated DNA Technologies (IDT) | N/A |
| Site directed mutagenesis on pTD195 to generate RTEL1ΔL834-G843, TD388 5′-ACCTCTTGGTCTTTGTCTAGC-3′ | Integrated DNA Technologies (IDT) | N/A |
| Site directed mutagenesis on pTD195 to generate RTEL1ΔL834-G843, TD389 5′-ACCAAGAGGTTCTCCAGGTGAAGAACAAGC-3′ | Integrated DNA Technologies (IDT) | N/A |
| Amplify backbone of GAL10 expression plasmid to clone POLE1 and POLE3_CBP. TD158 5′-GGCTGCAGGAATTCG-3′ | Integrated DNA Technologies (IDT) | N/A |
| Amplify backbone of GAL10 expression plasmid to clone POLE1 and POLE3_CBP. TD155 5′-TGTTTTATAACTAGTTATAGTTTTTTCTCCTTG-3′ | Integrated DNA Technologies (IDT) | N/A |
| Amplify backbone of GAL10 expression plasmid to clone POLE2 and POLE4. TD157 5′-TAAATTGAATTGAATTGAAATCGATAGATC-3′ | Integrated DNA Technologies (IDT) | N/A |
| Amplify backbone of GAL10 expression plasmid to clone POLE2 and POLE4. TD154 5′-TGTTTTATAGCGGCCGCTTATATTG-3′ | Integrated DNA Technologies (IDT) | N/A |
| Remove CBP tag from POLE3, TD303 5′-GTCGATAACTAATAAATTGAATTGAATTGAAATCGATAG-3′ | Integrated DNA Technologies (IDT) | N/A |
| Remove CBP tag from POLE3, TD302 5′-AATCACCATCATCATCCTTGTAGTCGAAACCTTGCAACTTGG-3′ | Integrated DNA Technologies (IDT) | N/A |
| Insert TEV_2XFLAG tag to POLE2, TD285 5′-TGATGGTGATTACAAGGATGACGACTAATAAATTGAATTGAATTGAAATCG-3′ | Integrated DNA Technologies (IDT) | N/A |
| Insert TEV_2XFLAG tag to POLE2, TD302 5′-AATCACCATCATCATCCTTGTAGTCGAAACCTTGCAACTTGG-3′ | Integrated DNA Technologies (IDT) | N/A |

| Reagent/Resource | Reference or Source | Identifier or Catalog number |
|---|---|---|
| **Chemicals, Enzymes and other reagents** | | |
| ORC | (Frigola et al, 2013) | N/A |
| Cdc6 | (Frigola et al, 2013) | N/A |
| Cdt1-Mcm2-7 | (Coster et al, 2014) | N/A |
| DDK | (On et al, 2014) | N/A |
| Sld3/7 | (Yeeles et al, 2015) | N/A |
| Cdc45 | (Yeeles et al, 2015) | N/A |
| Dpb11 | (Yeeles et al, 2015) | N/A |
| Sld2 | (Yeeles et al, 2015) | N/A |
| Pol ε | (Yeeles et al, 2015) | N/A |
| GINS | (Yeeles et al, 2015) | N/A |
| GINS (TwinStrep-Psf3) | (Lewis et al, 2022) | N/A |
| S-CDK | (Yeeles et al, 2015) | N/A |
| Mcm10 | (Yeeles et al, 2015) | N/A |
| Pol α | (Yeeles et al, 2015) | N/A |
| RPA | (Yeeles et al, 2015) | N/A |
| Ctf4 | (Yeeles et al, 2015) | N/A |
| Mrc1 | (Deegan et al, 2019) | N/A |
| Tof1-Csm3 | (Deegan et al, 2019) | N/A |
| RFC | (Yeeles et al, 2015) | N/A |
| PCNA | (Yeeles et al, 2015) | N/A |
| Pol δ | (Yeeles et al, 2015) | N/A |
| Top1 | (Deegan et al, 2019) | N/A |
| Fen1 | (Guillam et al, 2020) | N/A |
| Cdc9 | (Guillam et al, 2020) | N/A |
| Pif1 | (Deegan et al, 2019) | N/A |
| TopoIV | Ken Marians and Joe Yeeles | N/A |
| CMG | (Deegan et al, 2020) | N/A |

| Reagent/Resource | Reference or Source | Identifier or Catalog number |
|---|---|---|
| Rrm3 | (Deegan et al, 2019) | N/A |
| Rrm3ΔN | This study | N/A |
| Rrm3Δ111-130 | This study | N/A |
| Rrm3-2E | This study | N/A |
| Rrm3Δ86-110 | This study | N/A |
| Rrm3-CR | This study | N/A |
| Rrm3-6A | This study | N/A |
| BacPif1 | This study | N/A |
| Rrm3N-BacPif1 | This study | N/A |
| *hs*RTEL1 | This study | N/A |
| *hs*RTEL1ΔE813-E821 | This study | N/A |
| *hs*RTEL1ΔL834-G846 | This study | N/A |
| *hs*RTEL1ΔR875-V881 | This study | N/A |
| *hs*RTEL1ΔE813-E821 ΔR875-V881 | This study | N/A |
| *hs*CMG | This study | N/A |
| *hs*POL ε | This study | N/A |
| 3XFlag peptide | Sigma-Aldrich | F4799 |
| Roche cOmplete EDTA-free protease inhibitor cocktail | Roche | 000000011873580001 |
| Sigma protease inhibitor cocktail | Sigma-Aldrich | P8215 |
| Alpha factor | Pepceuticals | N/A |
| dNTPs | Promega | U1240 |
| NTPs | New England Biolabs | N0450L |
| [α-$^{32}$P]dCTP | Hartmann Analytics | FP-205 |
| [γ-$^{32}$P]ATP | Hartmann Analytics | FP-301 |
| Proteinase K | New England Biolabs | P8107S |
| Bovine Serum Albumin | Thermo Fisher | B14 |
| Phusion® High-Fidelity DNA Polymerase | New England Biolabs | M0530 |
| TaKaRa Ex Taq® DNA Polymerase | TaKaRa Bio | RR001A |
| XmaI | New England Biolabs | R0180S |
| BsiWI-HF | New England Biolabs | R3553S |
| SpeI-HF | New England Biolabs | R3133S |
| AscI | New England Biolabs | R0558S |
| RSM supplement mixture | Formedium | RSM0110 |
| β-glucuronidase | Sigma-Aldrich | G7770 |
| ECL western blotting detection reagent | Cytiva | RPN2106 |

| Reagent/Resource | Reference or Source | Identifier or Catalog number |
|---|---|---|
| **Software** | | |
| ChimeraX (v1.4) | UCSF Resource for Biocomputing, Visualization, and Informatics | https://www.cgl.ucsf.edu/chimerax/ |
| ImageJ (v1.53) | National Institute of Health | https://imagej.nih.gov/ij/ |
| Adobe Photoshop 2022 | Adobe | https://www.adobe.com/uk/products/photoshop.html |
| Adobe Illustrator 2022 | Adobe | https://www.adobe.com/uk/products/illustrator.html |
| AlphaFold (v2.0) | DeepMind | https://www.deepmind.com/open-source/alphafold |
| AlphaFold-multimer (v2.0) | DeepMind | https://github.com/deepmind/alphafold |
| ColabFold (v1.5.5) | Ovchinnikov & Steinegger laboratories | https://github.com/sokrypton/ColabFold |
| Colabfold Batch AlphaFold-2-multimer structure analysis pipeline | Ernst Schmid (Walter laboratory) | https://zenodo.org/records/8223143 |
| Epson Scan 3.9.3.4EN | Seiko Epson Corporation | https://www.epson.co.uk |
| **Other** | | |
| Anti-FLAG M2 affinity gel | Sigma-Aldrich | A2220 |
| Slide-A-Lyzer™ Dialysis Cassettes | Thermo Fisher | 66380 |
| illustra MicroSpin G-50 Columns | GE Healthcare | 27533002 |
| iBlot™ 2 Transfer Stacks | Invitrogen | IB23001 |
| YeaStar Genomic DNA Kit™ | Zymo research | D2002 |
| NuPAGE™ 4–12% Bis-Tris precast gels | Invitrogen | WG1402BX10 |
| 4–20% TBE gels | Invitrogen | EC62252BOX |
| Amersham hyperfilm ECL | Cytiva | 28906837 |
| Anti-FLAG® M2 Magnetic Beads | Sigma-Aldrich | M8823 |
| Dynabeads™ M-270 Epoxy | Invitrogen | 14301 |

Yeast strains constructed for tetrad dissection experiments were based on the W303 genetic background. Further details of strain construction can be found below. Full information regarding the genotypes of these all strains can be found in the Reagents and Tools Table.

## Protein expression and purification

*S. cerevisiae* ORC, Cdc6, Cdt1-Mcm2-7, DDK, S-CDK, Dpb11, GINS, Cdc45, Pol ε, Mcm10, CMG, RFC, PCNA, Top1, Pol α-primase, Sld3-7, Ctf4, RPA, Tof1-Csm3, Mrc1, Sld2, Pol δ, Fen1, Cdc9 ligase and Pif1 were expressed and purified as previously described (Coster et al, 2014; Deegan et al, 2019; Deegan et al, 2020;

Frigola et al, 2013; Guilliam and Yeeles, 2020; Lewis et al, 2022; On et al, 2014; Yeeles et al, 2015). *E. coli* TopoIV was a kind gift from Ken Marians and Joe Yeeles.

The strains in the Reagents and Tools Table were grown in YP + raffinose (2%) at 30 °C, to a density of $2$–$3 \times 10^7$ cells/ml. For ORC, Cdt1-Mcm2-7, Dpb11, Sld2 and Sld3-7 alpha factor mating pheromone (370 ng/mL) was added for 3 h at 30 °C to arrest cells in G1-phase. Galactose (2%) was added for 3 h at 30 °C to induce protein expression.

Cells were collected by centrifugation, washed once in that protein's corresponding lysis buffer without protease inhibitors and resuspended in 0.4 volumes of lysis buffer + protease inhibitors (Sigma-Aldrich, P8215 and Roche, 11836170001). The cell suspension was then frozen dropwise in liquid nitrogen. The resulting

yeast popcorn was crushed in a SPEX CertiPrep 6850 Freezer/Mill (3 × 2 min cycles, crushing rate 15) and the powder stored at −80 °C until required.

## Rrm3

45–90 g frozen cell powder (from 9 to 18 L culture) was thawed and resuspended in 260–390 mL of Rrm3 lysis buffer (50 mM Hepes-KOH pH 7.6, 0.02% NP-40, 10% glycerol, 1 mM DTT, 0.5 M KCl) + protease inhibitors (Sigma-Aldrich, P8215 and Roche, 11836170001). Insoluble material was removed by centrifugation ($235,000 \times g$, 4 °C, 1 h) and solid ammonium sulphate was gradually stirred into the soluble extract to 30% final concentration (10 min, 4 °C). Insoluble material was removed by centrifugation ($27,000 \times g$, 4 °C, 20 min) and the supernatant mixed with 1 mL anti-FLAG M2 affinity resin (Sigma-Aldrich) at 4 °C for 30 min.

The resin was collected in a disposable column and washed with 100 column volumes (CVs) of Rrm3 lysis buffer + protease inhibitors. The resin was then incubated with 10 CVs of Rrm3 lysis buffer + 5 mM MgOAc + 1 mM ATP for 10 min and then washed with a further 10 CVs of Rrm3 lysis buffer. Rrm3 was eluted by incubating the resin with 2 CVs of Rrm3 lysis buffer + 0.5 mg/mL 3FLAG peptide for 30 min followed by 1 CV of Rrm3 lysis buffer + 0.25 mg/mL 3FLAG peptide. The eluate was diluted to 0.3 M KCl by slow addition of saltless Rrm3 lysis buffer over 30 min, or by dialysing against buffer containing 25 mM Hepes-KOH pH 7.6, 0.02% NP-40, 10% glycerol, 1 mM DTT, 0.3 M KCl at 4 °C for 3 h. The resulting sample was loaded onto a 1 mL HiTrap heparin column equilibrated in 25 mM Hepes-KOH pH 7.6, 0.02% NP-40, 10% glycerol, 1 mM DTT, 0.3 M KCl, 10 mM MgOAc, 1 mM ATP.

Rrm3 was eluted with a 20 CV gradient from 0.3 to 1 M KCl in 25 mM Hepes-KOH pH 7.6, 0.02% NP-40, 10% glycerol, 1 mM DTT, 10 mM MgOAc, 1 mM ATP, unless the concentration of Rrm3 was low, in which case Rrm3 was eluted with a step to 0.5 M KCl in 25 mM Hepes-KOH pH 7.6, 0.02% NP-40, 10% glycerol, 1 mM DTT, 10 mM MgOAc, 1 mM ATP. Peak fractions containing Rrm3 were pooled and dialysed against 25 mM Hepes-KOH pH 7.6, 0.02% NP-40, 40% glycerol, 1 mM DTT, 0.35 M KCl at 4 °C overnight. The dialysed sample was recovered, aliquoted and snap frozen. Rrm3 mutants and truncations were purified with the same method as the wild-type protein.

## Rrm3N-BacPif1 and BacPif1

45–60 g frozen cell powder (9–12 L culture) was thawed and resuspended in 260 mL of Rrm3 lysis buffer + protease inhibitors (Sigma-Aldrich, P8215 and Roche, 11836170001). Insoluble material was removed by centrifugation ($235,000 \times g$, 4 °C, 1 h) and the soluble extract mixed with 1 mL anti-FLAG M2 affinity resin at 4 °C for 60 min. The resin was collected and washed with 100 CVs of Rrm3 lysis buffer + protease inhibitors. The resin was then incubated with 10 CVs of Rrm3 lysis buffer + 5 mM MgOAc + 1 mM ATP for 10 min and washed with 10 CVs of Rrm3 lysis buffer.

Rrm3N-BacPif1 was eluted by incubating the resin with 2 CVs of buffer Rrm3 lysis buffer + 0.5 mg/mL 3FLAG peptide for 30 min followed by 1 CV of Rrm3 lysis buffer + 0.25 mg/mL 3FLAG peptide. The eluate was diluted to 0.2 M KCl by slow addition of saltless Rrm3 lysis buffer over 30 mins. The resulting eluate was loaded onto a 1 mL MonoQ column equilibrated in 25 mM Hepes-KOH pH 7.6, 0.02% NP-40, 10% glycerol, 1 mM DTT, 0.2 M KCl, 10 mM MgOAc, 1 mM ATP. Rrm3N-BacPif1 was eluted with a 20 CV gradient from 0.2 to 1 M KCl in 25 mM Hepes-KOH pH 7.6, 0.02% NP-40, 10% glycerol, 1 mM DTT, 10 mM MgOAc, 1 mM ATP. Peak fractions were pooled and dialysed against 25 mM HEPES KOH pH 7.6, 40% glycerol, 0.02% NP-40, 1 mM DTT, 0.35 M KCl at 4 °C overnight. The dialysed sample was recovered, aliquoted and snap frozen. BacPif1 was purified in the same way as Rrm3N-BacPif1.

## hsCMG

110–120 g frozen cell powder (from 18 L culture) was thawed and resuspended in 260 mL of hsCMG lysis buffer (25 mM Hepes-KOH pH 7.6, 0.02% Tween-20, 10% glycerol, 0.3 M KCl, 2 mM MgOAc, 1 mM DTT) supplemented with protease inhibitors (Roche, 11836170001, 5 mM PMSF, 1 mM AEBSF, 1 µg/mL Pepstatin A). Insoluble material was removed by centrifugation ($235,000 \times g$, 4 °C, 1 h) and the supernatant then mixed with 6 mL IgG Sepharose 6 Fast Flow affinity resin (Cytiva) followed by rotation in a conical tube at 4 °C for 2 h.

The resin was pelleted at $500 \times g$ for 5 min, and the supernatant then re-incubated with an additional 6 mL IgG Sepharose 6 Fast Flow affinity resin (Cytiva) as described above. Resin recovered from both incubations was loaded in a disposable column and washed with 100 column volumes (CVs) of hsCMG lysis buffer + protease inhibitors. The resin was then resuspended and incubated with 10 CVs of hsCMG lysis buffer + 5 mM MgOAc + 1 mM ATP for 10 min on ice and then washed with a further 10 CVs of hsCMG lysis buffer. Elution was performed incubating the resin overnight with 300 µg of TEV protease (a kind gift from Dr. Axel Knebel) in hsCMG elution buffer (25 mM Hepes-KOH pH 7.6, 0.02% Tween-20, 10% glycerol, 0.2 M KCl, 2 mM MgOAc, 1 mM DTT) at 4 °C with rotation.

Eluate was collected and the resin was rinsed with a further 1 CV of hsCMG elution buffer. Both eluate and rinse fractions were pooled and loaded on a MonoQ 5/50 GL anion exchange column pre-equilibrated in hsCMG elution buffer. Elution was performed with a 30 CV gradient from 0.2 to 0.6 M KCl in hsCMG elution buffer. Peak fractions appearing at ~35 mS/cm conductivity were collected and dialysed against hsCMG storage buffer (25 mM Hepes-KOH pH 7.6, 0.02% Tween-20, 10% glycerol, 0.3 M KOAc, 2 mM MgOAc, 1 mM DTT). The dialysed sample was recovered, concentrated with an AMICON Ultra 15 30 KDa cutoff (Merck), aliquoted and snap frozen. Concentration was estimated by Bradford colorimetric assay.

## hsPOL ε

110–120 g frozen cell powder (from 18 L culture) was thawed and resuspended in 260 mL of hsPOL ε lysis buffer (25 mM Hepes-KOH pH 7.6, 0.02% NP-40, 10% glycerol, 0.1 M NaCl, 1 mM DTT) supplemented with protease inhibitors (Roche, 11836170001, 5 mM PMSF, 1 mM AEBSF, 1 µg/mL Pepstatin A). Insoluble material was removed by centrifugation ($235,000 \times g$, 4 °C, 1 h) and the supernatant mixed with 1 mL anti-FLAG M2 affinity resin (Sigma-Aldrich) with rotation in a conical tube at 4 °C for 1 h.

The resin was pelleted at $500 \times g$ for 5 min, and the supernatant incubated with an additional 1 mL anti-FLAG M2 affinity resin

(Sigma-Aldrich) as described above for CMG. Resin recovered from each incubation was loaded in a disposable column and washed with 100 column volumes (CVs) of hsPOL ε lysis buffer + protease inhibitors. The resin was then resuspended and incubated with 10 CVs of hsPOL ε lysis buffer + 5 mM MgOAc + 1 mM ATP for 10 min on ice and then washed with a further 10 CVs of hsPOL ε lysis buffer. Elution was performed by resuspending the resin with 4 CVs of hsPOL ε lysis buffer + 0.5 mg/mL 3FLAG peptide and incubating for 30 min with occasional agitation, followed by 4 CV of hsPOL ε lysis buffer + 0.25 mg/mL 3FLAG peptide. The 3FLAG tag was removed by adding 200 μg of TEV protease and rotating at 4 °C overnight.

Protein was then loaded on a MonoQ 5/50 GL anion exchange column pre-equilibrated in hsPOL ε lysis buffer. Protein was eluted with a 30 CV gradient from 0.1 to 0.8 M KCl in hsPOL ε lysis buffer. Peak fractions were concentrated with an AMICON Ultra 15 30 KDa cutoff (Merck), and loaded on a Superdex 200 Increase 10/300 GL column pre-equilibrated in hsPOL ε storage buffer (25 mM Hepes-KOH pH 7.6, 0.02% NP-40, 10% glycerol, 0.3 M KOAc, 1 mM DTT). Peak fractions were collected, concentrated as before, then aliquoted and snap frozen. Concentration was estimated by Bradford colorimetric assay.

## hsRTEL1

50–60 g frozen cell powder (from 9 L culture) was thawed and resuspended in 260 mL of RTEL1 lysis buffer (25 mM Hepes-KOH pH 7.9, 0.02% NP-40, 10% glycerol, 0.5 M NaCl, 0.5 mM TCEP) supplemented with protease inhibitors (Roche, 11836170001, 5 mM PMSF, 1 mM AEBSF, 1 μg/mL Pepstatin A). Insoluble material was removed by centrifugation (235,000 × $g$, 4 °C, 1 h) and the supernatant mixed with 0.5 mL anti-FLAG M2 affinity resin (Sigma-Aldrich) with rotation in a conical tube at 4 °C for 1 h.

The resin was pelleted at 500 × $g$ for 5 min, and the supernatant incubated with an additional 1 mL anti-FLAG M2 affinity resin (Sigma-Aldrich) as described above. Resin recovered from each incubation was loaded in a disposable column and washed with 100 column volumes (CVs) of RTEL1 lysis buffer + protease inhibitors. The resin was then resuspended and incubated with 10 CVs of RTEL1 lysis buffer + 5 mM MgOAc + 1 mM ATP for 10 min on ice and then washed with a further 10 CVs of RTEL1 lysis buffer. Elution was performed resuspending the resin with 1 CV of RTEL1 lysis buffer + 0.5 mg/mL 3FLAG peptide and incubating for 30 min with occasional agitation, followed by 1 CV of RTEL1 lysis buffer + 0.25 mg/mL 3FLAG peptide. The resulting eluates were pooled and loaded directly onto a 120 mL HiLoad Superdex 200 pg column pre-equilibrated in RTEL1 lysis buffer. Peak fractions were collected, pooled and dialysed against RTEL1 storage buffer (25 mM Hepes-KOH pH 7.9, 0.02% NP-40, 40% glycerol, 0.5 M NaCl, 0.5 mM TCEP) at 4 °C overnight. The dialysed sample was recovered, aliquoted and snap frozen. Concentration was estimated by running an SDS-PAGE gel with Coomassie staining against BSA dilution series. RTEL1 mutants were purified following the same method.

## DNA templates

The DNA templates pBS/ARS1WTA (3.2 kb) and pCFK1_WT (5.8 kb) have been described previously (Yeeles et al, 2015;

Marahrens and Stillman, 1992). Covalently closed plasmids for in vitro replication reactions were purified using alkaline lysis followed by caesium chloride density gradient centrifugation.

## Molecular weight markers

Molecular weight markers for native agarose gels were prepared by first dephosphorylating 17 μg λ DNA-HindIII Digest (New England Biolabs N3012S) with 10 U Antarctic Phosphatase (New England Biolabs M0289S) in total volume of 40 μL for 1 h at 37 °C. The phosphatase was inactivated by incubation at 80 °C for 10 min. 6.8 μg of dephosphorylated DNA was then labelled with γ-[$^{32}$P]-ATP using 40 units of T4 Polynucleotide Kinase (New England Biolabs M0201S) at 37 °C for 1 h, in a total reaction volume of 40 μL. Unincorporated γ-[$^{32}$P]-ATP was removed using Illustra MicroSpin G-50 columns (GE Healthcare) and 5 mM EDTA was added to the recovered sample.

## Reconstituted DNA replication reactions

Mcm2-7 loading and DDK phosphorylation was carried out by incubating 6 nM 3.2/5.8 kb plasmid DNA template (Reagents and Tools Table), 10 nM ORC, 20 nM Cdc6, 40 nM Cdt1·Mcm2-7, 20 nM DDK, 30 μM dATP-dCTP-dGTP-dTTP, 400 μM CTP-GTP-UTP and 33 nM α-[$^{32}$P]-dCTP in buffer containing 25 mM Hepes-KOH (pH 7.6), 100 mM KOAc, 0.02% NP-40-S, 0.1 mg/mL bovine serum albumin (BSA), 1 mM DTT, 10 mM Mg(OAc)$_2$, for 10 min at 30 °C.

DNA replication was initiated by addition of a protein mixture containing 30 nM S-CDK, 30 nM Dpb11, 20 nM GINS, 40 nM Cdc45, 30 nM Pol ε, 10 nM Mcm10, 10 nM RFC, 20 nM PCNA, 20 nM Top1, 20 nM Pol α-primase, 6.25 nM Sld3-7, 10 nM Ctf4, 50 nM RPA, 10 nM Tof1-Csm3, 20 nM Mrc1, 40 nM Sld2, 0.5 nM E. coli TopoIV, 0.25 nM Pol δ, 10 nM Fen1 and 20 nM Cdc9 ligase. Addition of the protein mixture diluted the MCM loading mix ~1.5–2-fold. For the experiments in Figs. 1C and EV2A, Pol δ, Fen1 and Cdc9 were omitted. In Fig. EV2A, Rrm3 and Pif1 were included at 12.5 and 5 nM, respectively. Reactions were incubated at 30 °C for 20 min.

For pulse-chase experiments (as in Figs. 1C, 2F and 5A,B), a cold chase of dATP-dCTP-dGTP-dTTP was added after the replication step to a final concentration of 600 μM, together with 5–20 nM Rrm3/Rrm3N-BacPif1/BacPif1 as indicated. The reactions (10–20 μL total volume) were incubated for a further 10 min at 30 °C and then stopped by addition of 25 mM EDTA, SDS (0.5%) and proteinase K (1/40 volumes) for 30 min at 37 °C. DNA was next purified by phenol/chloroform extraction. Unincorporated nucleotides were removed using Illustra MicroSpin G-50 columns, and the samples digested in 1x CutSmart buffer with 0.25 μL XmaI, SmaI or SpeI at 37 °C (or 25 °C for SmaI) for 30 min. Samples were then resolved in 0.8% horizontal native agarose gels in 1X TAE for ~16 h at 21 V. Gels were dried onto chromatography paper and exposed to BAS-MS Imaging Plates (Fujifilm), which were then developed on an FLA-5100 scanner (Fujifilm). Gels were subsequently exposed to Amersham Hyperfilm ECL (GE Healthcare) for presentation.

## Helicase assays

To prepare the helicase assay substrate for the experiments in Figs. EV1B, 2C and EV2B, 1 μM of a 47 nt PAGE-purified

oligonucleotide (TD254, Reagents and Tools Table) was labelled with γ-[$^{32}$P]-ATP using T4 Polynucleotide Kinase for 1 h at 37 °C in a 20 µL reaction volume. Unincorporated γ-[$^{32}$P]-ATP was removed using illustra MicroSpin G-50 columns. 1 µM labelled TD254 was annealed to 1 µM of complementary unlabelled oligonucleotide (TD255, Reagents and Tools Table; 25 bp complementary region and 22nt 5′ flap) in a 20 µL reaction containing 25 mM Hepes-KOH (pH 7.6), 100 mM NaCl and 5 mM Mg(OAc)$_2$. The reaction was heated to 95 °C for 5 min in a metal heating block, and then left to cool to room temperature for 3 h.

For helicase assays, 2 nM substrate was incubated with 5–25 nM helicase as indicated, in a 15 µL reaction volume containing 25 mM Hepes-KOH (pH 7.6), 0.1 mg/mL BSA, 2 mM Mg(OAc)$_2$ and 2 mM ATP. A complementary unlabelled 'trap' oligonucleotide (TD254, Reagents and Tools Table) was also included at 20 nM final concentration. Reactions were assembled on ice and then incubated at 30 °C for 30 min. Reactions were stopped by addition of 25 mM EDTA, 0.1% SDS, a 1/100 dilution of proteinase K and 5X native loading buffer (final concentration of 2% ficoll-400, 10 mM EDTA, 0.02% SDS, xylene cyanol). The samples were resolved in 4–20% TBE gels (Invitrogen EC62252BOX) at 200 V for 40 min in 1X TBE. Gels were placed on chromatography paper and exposed to BAS-MS Imaging Plates (Fujifilm), which were then developed on an FLA-5100 scanner (Fujifilm).

## Co-immunoprecipitation and immunoblotting

For anti-Sld5 immunoprecipitations (Figs. 2A,D,E and 4D), 1–10 nM Rrm3/Rrm3N-BacPif1/BacPif1 was incubated with 15 nM CMG and 30 nM Pol ε in buffer containing 25 mM Hepes-KOH (pH 7.6), 400 mM KOAc, 0.02% NP-40-S, 0.1 mg/mL BSA, 1 mM DTT and 10 mM Mg(OAc)$_2$, to give a 20 µL total reaction volume. The proteins were incubated on ice for 10 min to allow complex formation. 5 µL of each sample was then removed as input, added to 15 µL 1X SDS-PAGE sample loading buffer (Invitrogen NP0007) and boiled for 10 min at 75 °C. 5 µL of magnetic Dynabeads M-270 Epoxy (Invitrogen 14301) that had been coupled to anti-Sld5 antibodies (Reagents and Tools Table) were added to the remaining 15 µL sample and the reactions were incubated for 30 min at 4 °C, with mixing in a thermomixer (1400 rpm). Beads were then washed twice with 190 µL of buffer containing 25 mM Hepes-KOH (pH 7.6), 400 mM KOAc, 0.02% NP-40-S, 0.1 mg/mL BSA, 1 mM DTT and 10 mM Mg(OAc)$_2$. The bound proteins were eluted by addition of 20 µL 1X SDS-PAGE sample loading buffer and boiling for 10 min at 75 °C.

For anti-FLAG Rrm3 immunoprecipitations (Figs. 4B,C and EV3B,C), 10 nM Rrm3 was incubated with 15 nM GINS or Pol ε for 10 min on ice in buffer containing 25 mM Hepes-KOH (pH 7.6), 0.02% NP-40-S, 0.1 mg/mL BSA, 1 mM DTT and 10 mM Mg(OAc)$_2$, supplemented with 200 mM KOAc (GINS) or 400 mM KOAc (Pol ε). Reactions were performed as described above for anti-Sld5 IPs except 5 µL anti-FLAG M2 magnetic beads (Sigma-Aldrich M8823-1ML) were used instead of anti-Sld5 beads and washing was performed at 200 mM KOAc (GINS) or 400 mM KOAc (Pol ε).

For anti-FLAG RTEL1 immunoprecipitations (Figs. 7 and EV5), 10 nM RTEL1 was incubated with 15 nM hsCMG or hsPol ε for 10 min on ice in buffer containing 25 mM Hepes-KOH (pH 7.6), 0.02% NP-40-S, 0.1 mg/mL BSA, 1 mM DTT and 10 mM Mg(OAc)$_2$, supplemented with 100 mM KOAc (hsCMG) or

200 mM KOAc (hsPol ε). Reactions were performed as described above for anti-Sld5 IPs except 5 µL anti-FLAG M2 magnetic beads (Sigma-Aldrich M8823-1ML) were used instead of anti-Sld5 beads and washing was performed at 100 mM KOAc (hsCMG) or 200 mM KOAc (hsPol ε).

The samples were resolved by SDS-PAGE using 4–12% Bis-Tris gels (Invitrogen NP0322BOX) with NuPAGE MOPS SDS buffer (Invitrogen NP0001) at 200 V for 45 min. Proteins were then transferred onto a nitrocellulose iBlot membrane (Invitrogen IB301001) with the iBlot Dry Transfer System (Invitrogen). Membranes were blocked with 5% milk in TBST for 30 min at room temperature and then probed with primary antibodies (Reagents and Tools Table) overnight at 4 °C. The next day, membranes were washed 3X in TBST and incubated with horseradish peroxidase-conjugated secondary antibodies (Reagents and Tools Table) for 1 h. Membranes were washed again 3X in TBST, coated with ECL western blotting detection reagent (Cytiva RPN2106), and chemiluminescent signal was detected using Amersham Hyperfilm ECL (GE Healthcare).

## Yeast genetics

For the tetrad dissection experiments in Figs. 1E and EV1E, haploid strains carrying *rrm3ΔN* (or *rrm3Δ*) and *sml1Δ* alleles (yTDE11 and yTDE12—Reagents and Tools Table) were first generated by crossing strain KL14026 with strains yEH10 (*rrm3ΔN*) or yEH29 (*rrm3Δ*), followed by sporulation and tetrad dissection. The resultant haploids (yTDE11 and yTDE12) were then crossed with KL14026 to generate the yeast diploid strains yTDE13 and yTDE14 (Reagents and Tools Table).

To generate an *RRM3+/rrm3ΔN::URA3, DIA2/dia2Δ::HIS3* diploid (for the tetrad dissection experiment in Fig. 1D), the *rrm3ΔN* deletion was first introduced into the endogenous *RRM3* locus in the diploid *Saccharomyces cerevisiae* strain yKL1 (*MATa ade2-1 ura3-1 his3-11,15 trp1-1 leu2-3,112 can1-100/MATα ade2-1 ura3-1 his3-11,15 trp1-1 leu2-3,112 can1-100*) to generate an *RRM3+/ rrm3ΔN::URA3* heterozygous diploid (yEH9). To do this, a PCR product containing 50 bp of genomic DNA upstream of *RRM3* followed by *rrm3ΔN*, the URA3 marker and 50 bp genomic DNA downstream of *RRM3*, was transformed into yKL1. This PCR product was amplified from plasmid pTDE8 (Reagents and Tools Table) using primers TD301/TD284 (Reagents and Tools Table). Next, an *rrm3ΔN::URA3* haploid strain (yEH10) was derived by sporulation and tetrad dissection of yEH9. This was then crossed with a *dia2Δ::HIS3* haploid strain (yKL2714) to generate the diploid used in Fig. 1D.

rrm3-2E, -6A, -CR and Δ86-110 mutations were introduced into the endogenous RRM3 locus in the diploid S. cerevisiae strain yKL2713 (MATa ade2-1 ura3-1 his3-11,15 trp1-1 leu2-3,112 can1-100/MATα ade2-1 ura3-1 his3-11,15 trp1-1 leu2-3,112 can1-100 dia2::HIS3/DIA2) to generate heterozygous diploids yOO35, yOO28, yOO29 and yOO24 (for tetrad dissections in Figs. 5C,D and EV4C,D). To do this, PCR products containing 1 kb of genomic DNA upstream of RRM3 followed by rrm3-2E, -6A, -CR or Δ86-110, the URA3 marker and 50 bp genomic DNA downstream of RRM3, were transformed into KL2713. The PCR products containing rrm3-2E, -6A, -CR and Δ86-110 mutations were amplified from plasmids pOO24-pOO27 (Reagents and Tools Table) using primers TD284/TD371 (Reagents and Tools Table).

Clones were selected by plating on -URA selective plates, and those with integration into one copy of *RRM3* were identified by PCR amplification (TD306/TD307; Reagents and Tools Table) following genomic DNA extraction. *RRM3* mutations were confirmed by DNA sequencing.

For tetrad dissection, diploid yeast strains (Reagents and Tools Table) were patched onto RSM plates (2% agar, 1.5% KOAc, 0.25% yeast extract, 0.077% RSM supplement (ForMedium), 0.1% glucose) and incubated for 2 days at 30 °C. Sporulated cells were picked with a pipette tip and resuspended in 150 μL water. 10 μL β-Glucuronidase (Sigma-Aldrich, G7770) were added and the cells incubated at room temperature for 15–20 min. 10 μL were then streaked onto YPD plates and tetrads were dissected using a micromanipulator (MSM400, Singer Instruments). The cells were grown for 2 days at 30 °C before being imaged. Spores from each tetrad were genotyped by replica plating onto appropriate selective plates.

## AlphaFold-Multimer

AlphaFold-Multimer (AlphaFold2_multimer_v2) was run on ColabFold (ColabFold v1.5.5) to predict pairwise interactions between full-length *S. cerevisiae* Rrm3 and the following *S. cerevisiae* proteins (all full-length): GINS tetramer (Sld5, Psf1-3), Mcm2, Mcm3, Mcm4, Mcm5, Mcm6, Mcm7, Cdc45, Ctf4, Mcm10, Mrc1, PCNA monomer Pol30, Pol α subunits Pol1, Pol12, Pri1-Pri2, Pol ε subunits Dpb2, Dpb3-4, Pol2, RPA subunits Rfa1-Rfa2-Rfa3 and Tof1-Csm3. The following parameters were used: num_models=5, num_recycles=3, num_relax=0, template_mode=pdb100, except for GINS, for which num_relax=1 was used. Colabfold sequence alignments were performed using Mmseq2.

Pairwise interactions of *Homo sapiens* RTEL1 isoform 1 (NP_057518.1) with POLE2 and the GINS tetramer (SLD5, PSF1-3) were modelled using the same parameters, except AlphaFold2_multimer_v3 was used for GINS. The same parameters were used for the modelling of *Homo sapiens* PIF1, FANCJ, DDX3, DDX11, HELB and XPD with POLE2 and GINS.

To analyse the predictions produced by AlphaFold-Multimer (Appendix Tables S1, S2), we used an analysis pipeline developed by the Walter laboratory (Lim et al, 2023; Schmid, 2023). Definitions of the various metrics generated by these analyses are given in Appendix Tables S1, 2 and Lim et al, 2023; Schmid, 2023.

## Structural modelling in ChimeraX

For Fig. 1A, the PDB files corresponding to AlphaFold-predicted structures of *S. cerevisiae* Rrm3 and BacPif1 (from *Bacteroides* sp 2 1 16) were loaded in ChimeraX. The matchmaker tool was then used to align these two proteins based on the helicase domain of Rrm3 (residues 230–723).

To assemble the model in Fig. 3D, the PDB file 7PMK was loaded in ChimeraX and the following proteins were deleted: Ctf4, Dia2, Skp1 and Tof1-Csm3. The matchmaker tool was then used to align the Rrm3-Dpb2 AlphaFold-Multimer model to Dpb2 in the 7PMK structure, and all residues were deleted except Rrm3 residues 86–110. The Rrm3-GINS AlphaFold-Multimer model was then aligned to Sld5 in the 7PMK structure, and all residues deleted except Rrm3 residues 114–122.

To assemble the model in Fig. 6D, the PDB file 7PLO was loaded in ChimeraX and the following proteins were deleted: CTF4,

TIMELESS, TIPIN, LRR1, ELOB, CLASPIN, ELOC, CUL2 and RBX1. The matchmaker tool was then used to align the RTEL1-POLE2 AlphaFold-Multimer model to POLE2 in the 7PLO structure, and all residues were deleted except RTEL1 residues 834–846. The RTEL1-GINS AlphaFold-Multimer model was then aligned to PSF1 in the 7PLO structure, and all residues deleted except RTEL1 residues 813–820 and 876–880.

## Quantification and statistical analysis

To quantify the percentage of full-length products in reconstituted DNA replication reactions, 16-bit tiff files of gel images were opened in ImageJ. Boxes were drawn around each sample lane, and peaks corresponding to Late Replication Intermediates and full-length products were selected manually for each sample. The percentage of full-length products was calculated as a percentage of the total replication products in each lane. The same process was performed for helicase assay gels, except peaks were selected corresponding to the annealed substrate and unwound product. The experiments in Figs. EV1B, 2C,F and 5A,B were carried out three times and the mean and standard deviation values for each are plotted in Figs. EV1C, EV2B,C, EV4A,B, respectively.

## Data availability

This study includes no data deposited in external repositories.

The source data of this paper are collected in the following database record: biostudies:S-SCDT-10_1038-S44318-024-00168-4.

## Peer review information

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

## Acknowledgements

We gratefully acknowledge the support of the Medical Research Council (core grant MC_UU_00035/4 to TD). We thank Joe Yeeles for helpful discussion and comments on the manuscript, Ken Marians for *E. coli* TopoIV, Ryo Fujisawa for providing beads coupled to anti-Sld5 antibody, Jacob Lewis and Alessandro Costa for sharing the expression plasmid for budding yeast GINS, Ernst Schmid for help with Python scripts for AlphaFold-Multimer analyses, and Karim Labib and Cristian Polo Rivera for sharing unpublished information and reagents.

## Author contributions

**Ottavia Olson**: Conceptualization; Investigation. **Simone Pelliciari**: Investigation. **Emma D Heron**: Investigation. **Tom D Deegan**: Conceptualization; Supervision; Funding acquisition; Investigation; Writing—original draft; Project administration; Writing—review and editing.

Source data underlying figure panels in this paper may have individual authorship assigned. Where available, figure panel/source data authorship is listed in the following database record: biostudies:S-SCDT-10_1038-S44318-024-00168-4.

## Disclosure and competing interests statement

The authors declare no competing interests.

# Expanded View Figures

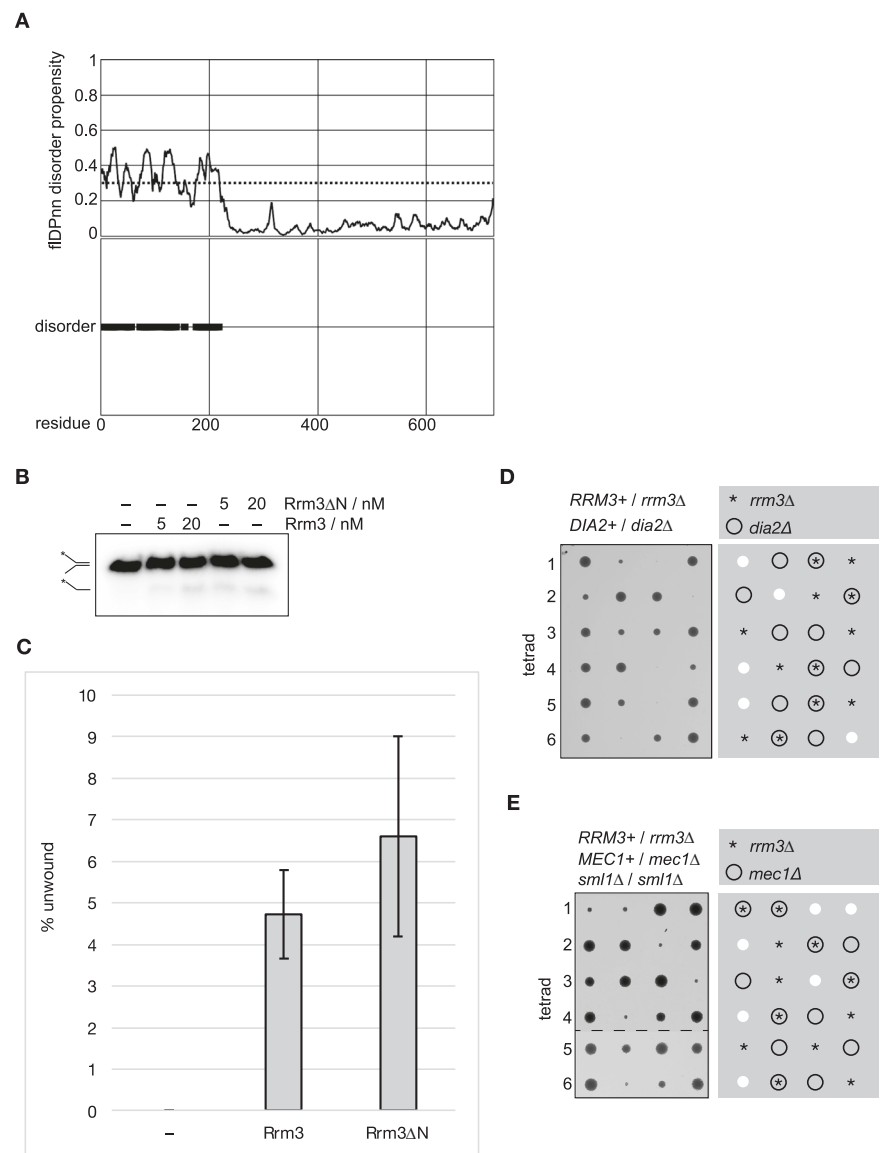

**Figure EV1. Characterisation of *rrm3ΔN* mutant in vitro and in vivo.**

(A) Disorder prediction for *S. cerevisiae* Rrm3, generated using the flDPnn webserver. Residue numbers are given on the x-axis. (B) The ability of Rrm3 and Rrm3ΔN to unwind a 25 bp DNA duplex, formed by annealing oligonucleotide TD254 to TD255, was monitored as described in Methods. * indicates [32]P-labelling of TD254. (C) Similar experiments to (B) were performed three times. The percentage of unwound product was quantified in each case for reactions containing 5 nM of Rrm3, and the figure presents the mean values with standard deviations. (D, E) Diploid yeast cells of the indicated genotypes were sporulated and the resulting tetrads were then dissected and grown on YPD medium for 2 days at 30 °C.

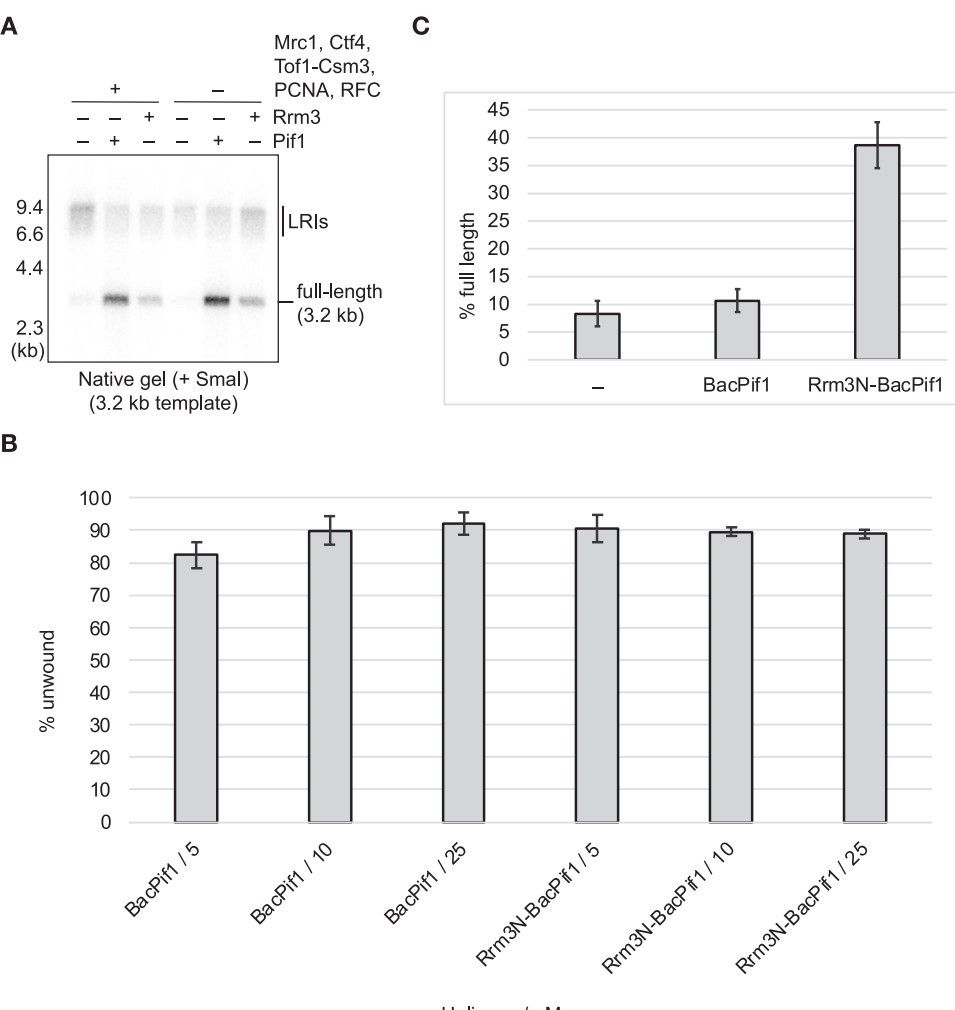

**Figure EV2. Supporting data for Fig. 2.**

(A) A 3189 bp plasmid template (pBS/ARS1WTA) was replicated in the presence or absence of Rrm3 (12.5 nM) or Pif1 (5 nM) and the indicated replisome components. SmaI-digested radiolabelled replication products were resolved in a native agarose gel and detected by autoradiography. **(B)** Similar experiments to Fig. 2C were performed three times. The percentage of unwound product was quantified in each case, and the figure presents the mean values with standard deviations. **(C)** Similar experiments to Fig. 2F were performed three times. The percentage full-length products was quantified in each case, and the figure presents the mean values with standard deviations. Quantification was performed for BacPif1 and Rrm3N-BacPif1 samples that included 5 nM of each helicase.

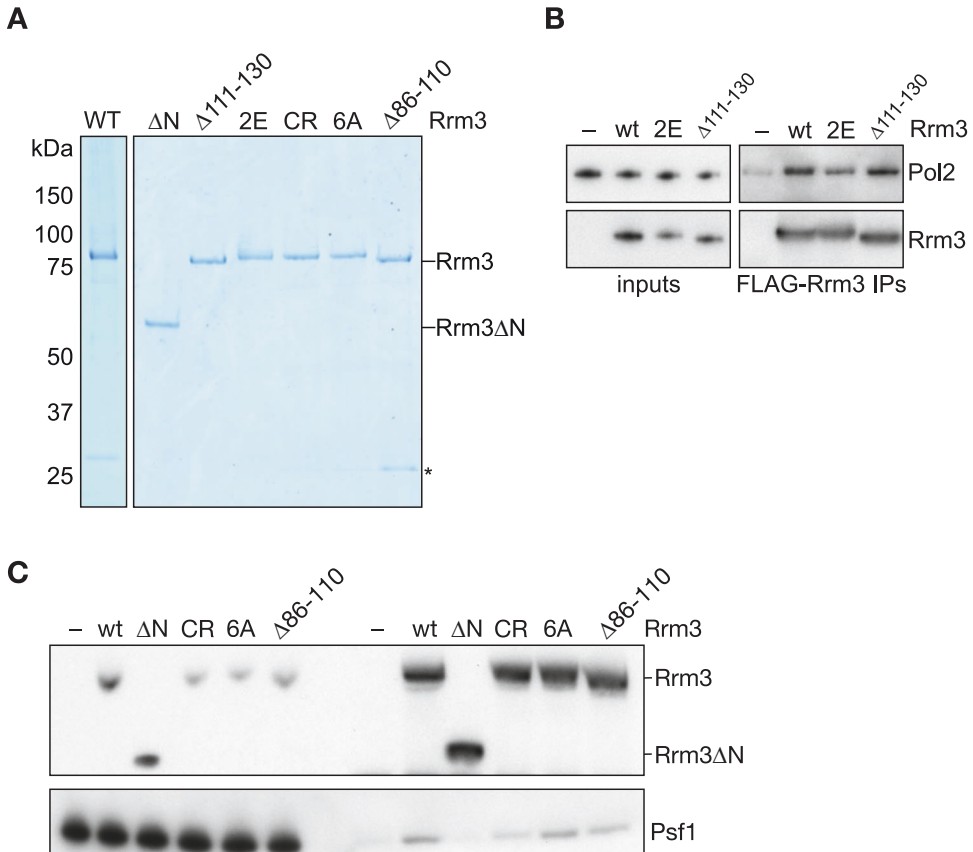

**Figure EV3. Generation and characterisation of CMGE-binding mutants of Rrm3.**

(A) Purified wild-type or mutant versions of Rrm3 visualised by SDS-PAGE and Coomassie staining. * is a contaminating protein. (B, C) Purified Polε (B) or tetrameric GINS complex (C) were mixed with FLAG-tagged wild-type Rrm3 or the indicated Rrm3 mutants. Resultant complexes were isolated by anti-FLAG immunoprecipitation and detected by SDS-PAGE and immunoblotting. Rrm3 was detected by anti-FLAG immunoblotting.

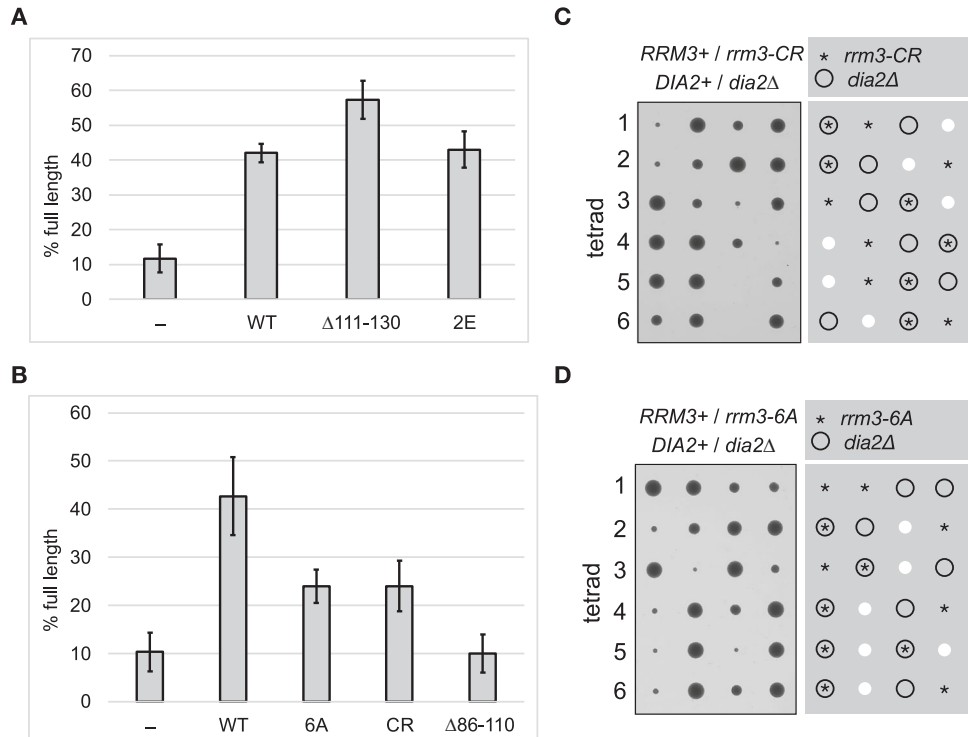

**Figure EV4. Supporting data showing that CMGE binding is critical for Rrm3 function.**

(A, B) Similar experiments to Fig. 5A (**A**) and 5B (**B**) were performed three times. The percentage full-length products was quantified in each case, and the figure presents the mean values with standard deviations. (C, D) Diploid yeast cells of the indicated genotypes were sporulated and the resulting tetrads were then dissected and grown on YPD medium for 2 days at 30 °C.

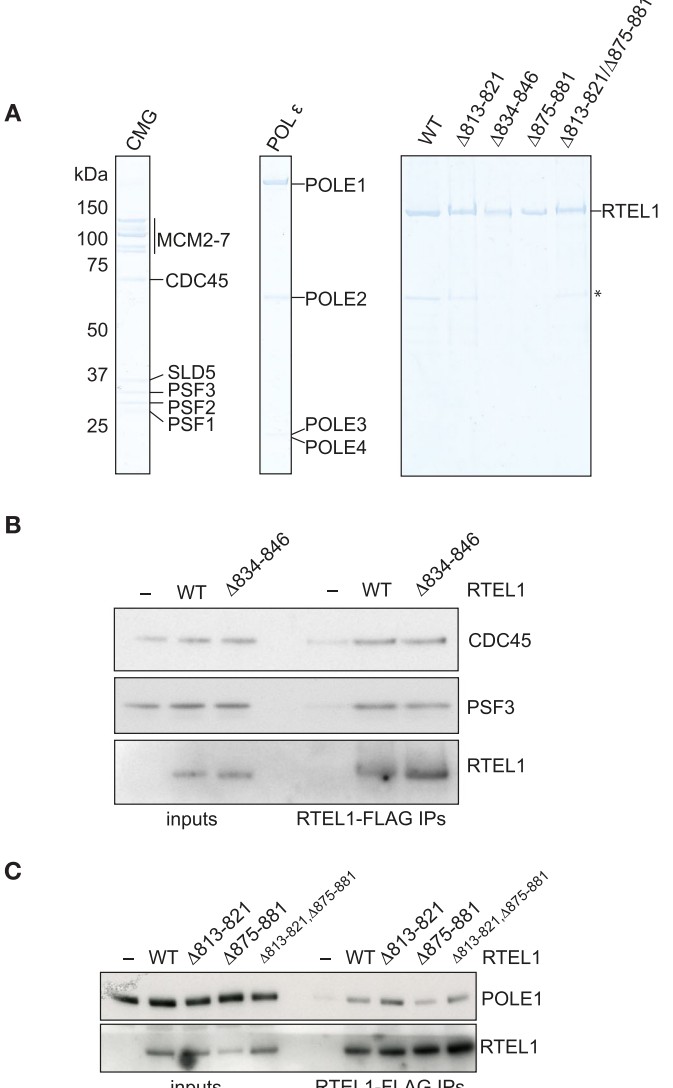

**Figure EV5. Generation and characterisation of CMGE-binding mutants of RTEL1.**

(A) Wild type or mutant versions of *Homo sapiens* RTEL1, CMG and POL ε purified after expression in budding yeast and visualised by SDS-PAGE and Coomassie staining. * indicates a contaminant in purified RTEL1. (B, C) Purified CMG (B) or POL ε (C) were mixed with FLAG-tagged wild-type RTEL1 or the indicated RTEL1 mutants. Resultant complexes were isolated by anti-FLAG immunoprecipitation and detected by SDS-PAGE and immunoblotting. RTEL1 was detected by anti-FLAG immunoblotting.

