## [Peer Review File · The EMBO Journal]

A common mechanism for recruiting the Rrm3 and RTEL1 accessory helicases to the eukaryotic replisome

Tom Deegan, Ottavia Olson, Simone Pellicciari, and Emma Heron

Corresponding author(s): Tom Deegan (t.deegan@ed.ac.uk)

Review Timeline:

Submission Date:	21st Mar 24
Editorial Decision:	10th Apr 24
Revision Received:	4th Jun 24
Editorial Decision:	24th Jun 24
Revision Received:	24th Jun 24
Accepted:	26th Jun 24

Editors: Hartmut Vodermaier and Cornelius Schneider

Transaction Report:

Dr. Tom D. Deegan
University of Edinburgh
United Kingdom

10th Apr 2024

Re: EMBOJ-2024-117324
A mechanism for recruiting accessory DNA helicases to the eukaryotic replisome

Dear Tom,

Thank you for submitting your study on replisome recruitment of accessory helicases to The EMBO Journal. I sent it to three expert referees, who have now returned the reports copied below. Since they all appreciate the importance of the question and the straightforward new insights into it we would be happy to consider the study further for publication. At the same time, referee 3 notes that the functional characterization of the observed interactions remains somewhat incomplete; and while I feel that testing this for human CMG/RTEL/POLE may fall beyond the scope of this work, I do agree that addressing some of the new Rrm3 functions with cellular assays in yeast would indeed further strengthen the study.

I am therefore inviting you to prepare a revised manuscript addressing this main issue, as well as the few more specific queries raised by referees 1 and 3. Please be reminded that it is our policy to allow only a single round of (major) revision, making it important to carefully respond to all points raised at this stage; please do not hesitate to contact me already during the early stages of your revision work in case you would like to discuss any of the issues raised by the reviewers, or if you should require an extension of the revision period.

Detailed information on preparing, formatting and uploading a revised manuscript can be found below and in our Guide to Authors. Thank you again for the opportunity to consider this work for The EMBO Journal, and I look forward to your revision in due time.

With kind regards,

Hartmut

9) Digital image enhancement is acceptable practice, as long as it accurately represents the original data and conforms to community standards. If a figure has been subjected to significant electronic manipulation, this must be clearly noted in the figure legend and/or the 'Materials and Methods' section. The editors reserve the right to request original versions of figures and the original images that were used to assemble the figure. Finally, we generally encourage uploading of numerical as well as gel/blot image source data; for details see: embopress.org/page/journal/14602075/authorguide#sourcedata

At EMBO Press, we ask authors to provide source data for the main manuscript figures. Our source data coordinator will contact you to discuss which figure panels we would need source data for and will also provide you with helpful tips on how to upload and organize the files.

In the interest of ensuring the conceptual advance provided by the work, we recommend submitting a revision within 3 months (9th Jul 2024). Please discuss the revision progress ahead of this time with the editor if you require more time to complete the revisions. Use the link below to submit your revision:

Link Not Available

Referee #1:

In this manuscript by Olson et al., the authors investigate how accessory helicases interact with the replisome in eukaryotes. Most of the work focused on the well established *Saccharomyces cerevisiae* system, where the PIF1 family helicase Rrm3 aids the replisome in DNA replication through sites of non-histone protein-DNA complexes. Using a combination of biochemistry, genetics, and AlphaFold modelling, the authors demonstrate that the natively disordered N-terminal domain of Rrm3 contains two short linear interaction motifs (SLIMs) that enable physical association of Rrm3 with one subunit each of Pol-epsilon and the GINS complex. Taking an analogous approach, the authors also demonstrate that RTEL1 similarly interacts with the replisome in metazoans. In both cases, these interactions are functionally important.

Overall, this was an insightful and straightforward piece of science that will be of interest to the DNA replication and genome integrity field. I only have a few minor concerns that I believe should be addressed before the manuscript is suitable for publication:

1) AlphaFold predicts that Rrm3 has an ~230-aa N-terminal IDR, yet when the authors investigate the importance of this IDR, they only truncated the first 193 aa (see lines 156-159). Why was this shorter truncation chosen rather than deleting the first 230 aa? Justifying this choice is important because the Bochman, Byrd/Raney, and Zakian labs have all published Rrm3 "full" N-terminal truncations that nevertheless differ in where the truncation was made. I don't suggest that the authors test any of these other truncation variants - their data are clear - but some mention should be made about why this particular truncation mutant was chosen.

2) Line 204 states that the Rrm3N-BacPif1 fusion has comparable activity to BacPif1, but Figure 2C is of low quality and unconvincing. Multiple groups have shown vigorous unwinding by bacterial PIF1 helicases, which does not appear to be the case with the anemic unwinding shown in Figure 2C. Further, the Rrm3N-BacPif1 activity appears to decrease with increasing

concentration. A better gel image is warranted, and graphed averages from triplicate data (+/- standard deviation) would be appropriate as well.

3) In the Methods section, many of the protein purification sub-sections begin by stating the volume of cell culture used (e.g., 9-18 L for Rrm3). While this is appreciated because many methods sections lack even that amount of detail, a more useful piece of information would be the mass of cell pellet collected from culture (e.g., 50 g). Depending on the expression host and culturing technique used (e.g., shaking flasks vs. fermenters or log-phase cells vs. high-density autoinduction), a given volume of cell culture could produce wildly different amounts of biomass to process.

Referee #2:

In this manuscript, Olson et al use a combination of structural modeling, biochemical and genetic experiments to elucidate the mechanism by which the accessory helicase Rrm3 is recruited to the budding yeast replisome: the authors extend these findings to RTEL1, which performs analogous functions in metazoans, by demonstrating a similar mode of binding. The question of how accessory helicases are recruited to support DNA replication past hard-to-replicate sites is a very interesting one, and the experiments presented here provide strong support for a novel model.

I have only good things to say about this manuscript: the structural modeling is interesting; the experiments are surgically targeted and the resulting data are of a very high-quality; the manuscript is clearly written. The findings are exciting and will be of significant interest to groups working in the fields of DNA replication and genome integrity. Instead of nitpicking some very minor points, I will simply encourage publication of this excellent manuscript in its current form.

Referee #3:

Uninterrupted replication fork progression across the genome is facilitated by accessory DNA helicases that mediate replisome bypass of physical obstacles on the DNA. How the replicative DNA helicase in eukaryotes, CMG, and accessory DNA helicases are physically coordinated is incompletely understood. In this study, Olson et al. utilize AlphaFold multimer to identify binding sites for yeast Rrm3 and human RTEL1 within yeast or human CMG, respectively. Interestingly, both Rrm3 and RTEL1 feature unstructured domains that harbor sequential short linear interaction motifs (SLIMs) interacting with GINS or Pol-epsilon, respectively, indicating a conserved mode of interaction for accessory helicases in yeast and humans. The predicted binding sites in Rrm3 and RTEL1 are verified biochemically in pull-down assays with purified proteins while the functional significance of the interaction motifs in Rrm3 are also verified using *in vitro* DNA replication assays and genetic synthetic interactions with DIA2 deletions in yeast cells.

Overall, this paper proposes a plausible model for the positioning of accessory 5'-3' helicases at eukaryotic replisomes that will be of interest to researchers in the DNA replication field. The data are generally of high quality and the experimental design is conceptually clear and straightforward.

The main weakness of the paper is the incomplete functional characterization of the observed interactions. For example:

- With regard to the replication termination function of Rrm3, cellular replication termination assays (as the authors have done in their 2019 Molecular Cell paper) could help strengthen the conclusions based on the *in vitro* termination assays reported here.
- An *in vivo* assessment of the termination function of Rrm3 mutations identified here may also help address why mutation of the GINS interaction motif in Rrm3 does not affect the replication termination function of Rrm3 *in vitro* yet exhibits a synthetic growth defect with DIA2 deletion in cells. Perhaps conditions are simply more stringent *in vivo* than *in vitro* and loss of the GINS interaction will exhibit a termination defect *in vivo*? This is relevant as the authors propose a similar or identical mode of action for Rrm3 at replisomes in both scenarios.
- The functional significance of the reported RTEL1 interactions with human CMG and Pol-epsilon remains untested altogether.

Additional points:

- DNA unwinding (helicase) assays and *in vitro* termination assays should be performed in multiple repeats and the data quantified and plotted to allow assessment of the experimental variability and statistical significance of the observed effects.
- A number of additional factors have been reported previously to mediate the recruitment of RTEL1 to sites of DNA replication, including TRF2 (PMID: 29727617), Poldip3 (PMCID: PMC7397856) and SLX4 (PMID 32398829). These reports should be discussed in light of the direct RTEL1-CMGE interaction proposed here.
- Figure EV2A: The data suggest that Pif1 is more proficient in mediating replication termination than Rrm3. Does Pif1 contain a disordered domain with predicted SLIMs similar to Rrm3? Or is it possible that accessory helicases can utilize distinct modes of coordination with the replisome as opposed to only the conserved interaction proposed here?

Dear Tom,

Thank you for submitting your study on replisome recruitment of accessory helicases to The EMBO Journal. I sent it to three expert referees, who have now returned the reports copied below. Since they all appreciate the importance of the question and the straightforward new insights into it we would be happy to consider the study further for publication. At the same time, referee 3 notes that the functional characterization of the observed interactions remains somewhat incomplete; and while I feel that testing this for human CMG/RTEL/POLE may fall beyond the scope of this work, I do agree that addressing some of the new Rrm3 functions with cellular assays in yeast would indeed further strengthen the study.

I am therefore inviting you to prepare a revised manuscript addressing this main issue, as well as the few more specific queries raised by referees 1 and 3. Please be reminded that it is our policy to allow only a single round of (major) revision, making it important to carefully respond to all points raised at this stage; please do not hesitate to contact me already during the early stages of your revision work in case you would like to discuss any of the issues raised by the reviewers, or if you should require an extension of the revision period.

Detailed information on preparing, formatting and uploading a revised manuscript can be found below and in our Guide to Authors. Thank you again for the opportunity to consider this work for The EMBO Journal, and I look forward to your revision in due time.

With kind regards,

Hartmut

Referee #1:

In this manuscript by Olson et al., the authors investigate how accessory helicases interact with the replisome in eukaryotes. Most of the work focused on the well established *Saccharomyces cerevisiae* system, where the PIF1 family helicase Rrm3 aids the replisome in DNA replication through sites of non-histone protein-DNA complexes. Using a combination of biochemistry, genetics, and AlphaFold modelling, the authors demonstrate that the natively disordered N-terminal domain of Rrm3 contains two short linear interaction motifs (SLIMs) that enable physical association of Rrm3 with one subunit each of Pol-epsilon and the GINS complex. Taking an analogous approach, the authors also demonstrate that RTEL1 similarly interacts with the replisome in metazoans. In both cases, these interactions are functionally important.

Overall, this was an insightful and straightforward piece of science that will be of interest to the DNA replication and genome integrity field. I only have a few minor concerns that I believe should be addressed before the manuscript is suitable for publication:

We thank the reviewer for their appreciation of our work and their helpful suggestions that follow.

1) AlphaFold predicts that Rrm3 has an ~230-aa N-terminal IDR, yet when the authors investigate the importance of this IDR, they only truncated the first 193 aa (see lines 156-159). Why was this shorter truncation chosen rather than deleting the first 230 aa? Justifying this choice is important because the Bochman, Byrd/Raney, and Zakian labs have all published Rrm3 "full" N-terminal truncations that nevertheless differ in where the truncation was made. I don't suggest that the authors test any of these other truncation variants - their data are clear - but some mention should be made about why this particular truncation mutant was chosen.

Our original design of the Rrm Δ N protein came before the development of AlphaFold, and was therefore guided by the protein disorder and secondary structure prediction tools available at the time, as well as published data from the Zakian group (e.g. PMID: 15579680), which suggested that deletion of the first 193 residues of Rrm3 resulted in a loss-of-function phenotype in vivo. Once AlphaFold was released, we realised that the Rrm3 IDR was in fact predicted to extend to residue 230. However, we have not since gone on to test any additional truncations of the Rrm3 IDR, as removal of the first 193 amino acids (which contain the Sld5- and Dpb2-interacting SLIMs that mediate Rrm3 recruitment) is already sufficient to completely ablate Rrm3 binding to CMGE, and Rrm3 function. We have now referenced PMID: 15579680 on lines 156-157 of our revised manuscript, to help justify our design of the Rrm Δ N protein.

2) Line 204 states that the Rrm3N-BacPif1 fusion has comparable activity to BacPif1, but Figure 2C is of low quality and unconvincing. Multiple groups have shown vigorous unwinding by bacterial PIF1 helicases, which does not appear to be the case with the anemic unwinding shown in Figure 2C. Further, the Rrm3N-BacPif1 activity appears to decrease with increasing concentration. A better gel image is warranted, and graphed averages from triplicate data (+/- standard deviation) would be appropriate as well.

Following the reviewer's suggestion, we have now repeated this experiment three times (Fig. 2C and EV2B). The inclusion of a 'trap' oligonucleotide in the new experiments results in enhanced DNA unwinding by both BacPif1 and Rrm3N-BacPif1, indicating that the lower levels of helicase activity presented in the original Fig. 2C were very likely a result of the re-annealing of substrate DNA strands after unwinding. Our new data supports our conclusion that the helicase activities of Rrm3N-BacPif1 and BacPif1 are comparable across a range of protein concentrations.

3) In the Methods section, many of the protein purification sub-sections begin by stating the volume of cell culture used (e.g., 9-18 L for Rrm3). While this is appreciated because many methods sections lack even that amount of detail, a more useful piece of information would be the mass of cell pellet collected from culture (e.g., 50 g). Depending on the expression host and culturing technique used (e.g., shaking flasks vs. fermenters or log-phase cells vs. high-density autoinduction), a given volume of cell culture could produce wildly different amounts of biomass to process.

We agree that this information is useful for the reader and have now added it to the Methods section.

Referee #2:

In this manuscript, Olson et al use a combination of structural modeling, biochemical and genetic experiments to elucidate the mechanism by which the accessory helicase Rrm3 is recruited to the budding yeast replisome: the authors extend these findings to RTEL1, which performs analogous functions in metazoans, by demonstrating a similar mode of binding. The question of how accessory helicases are recruited to support DNA replication past hard-to-replicate sites is a very interesting one, and the experiments presented here provide strong support for a novel model.

I have only good things to say about this manuscript: the structural modeling is interesting; the experiments are surgically targeted and the resulting data are of a very high-quality; the manuscript is clearly written. The findings are exciting and will be of significant interest to groups working in the fields of DNA replication and genome integrity. Instead of nitpicking some very minor points, I will simply encourage publication of this excellent manuscript in its current form.

We thank the reviewer for their appreciation of our work.

Referee #3:

Uninterrupted replication fork progression across the genome is facilitated by accessory DNA helicases that mediate replisome bypass of physical obstacles on the DNA. How the replicative DNA helicase in eukaryotes, CMG, and accessory DNA helicases are physically coordinated is incompletely understood. In this study, Olson et al. utilize AlphaFold multimer to identify binding sites for yeast Rrm3 and human RTEL1 within yeast or human CMG, respectively. Interestingly, both Rrm3 and RTEL1 feature unstructured domains that harbor sequential short linear interaction motifs (SLIMs) interacting with GINS or Pol-epsilon, respectively, indicating a conserved mode of interaction for accessory helicases in yeast and humans. The predicted binding sites in Rrm3 and RTEL1 are verified biochemically in pull-down assays with purified proteins while the functional significance of the interaction motifs in Rrm3 are also verified using in vitro DNA replication assays and genetic synthetic interactions with DIA2 deletions in yeast cells.

Overall, this paper proposes a plausible model for the positioning of accessory 5'-3' helicases at eukaryotic replisomes that will be of interest to researchers in the DNA replication field. The data are generally of high quality and the experimental design is conceptually clear and straightforward.

We thank the reviewer for their appreciation of our work and constructive comments below.

The main weakness of the paper is the incomplete functional characterization of the observed interactions. For example:

- With regard to the replication termination function of Rrm3, cellular replication termination assays (as the authors have done in their 2019 Molecular Cell paper) could help strengthen the conclusions based on the in vitro termination assays reported here.

This is a good suggestion, and we agree that assessing our Rrm3 mutants in in vivo replication termination assays would add to our already extensive characterisation of the Rrm3-CMGE interactions, using AlphaFold modelling, protein-protein interaction assays, yeast genetics and in vitro DNA replication termination assays. Notably, the in vivo plasmid replication experiments that the reviewer refers to are in fact more challenging and time-consuming than one might expect. Specifically, to resolve the Late Replication Intermediates (which result from the stalling of converging replication forks in the absence of Rrm3) from other plasmid isoforms, we have to run native agarose gels very slowly, for at least 10 days, meaning that each experiment takes around 3 weeks from start to finish.

We have made multiple attempts at such experiments using our rrm3 mutant yeast strains, in collaboration with Jon Baxter (University of Sussex), a world-leading expert in the analysis of plasmid DNA replication termination in budding yeast (e.g. see PMID: 30850330, 18570880). Unfortunately, Jon's lab have experienced a number of challenging technical issues with re-establishing this assay, having not performed such experiments in the last couple of years. Specifically, key lab reagents required for these experiments, such as the AP-conjugated anti-fluorescein antibody and substrate used for Southern blotting, have gone out of date, and failed to work properly in our experiments. Frustratingly, despite enquiring with multiple suppliers, we have been told that we will have to wait until September (at the earliest) to get new batches of some reagents. Accordingly, any further attempts at these experiments would result in an extensive delay to resubmission. In light of these extenuating circumstances, as well as our existing genetic data showing synthetic sickness between our rrm3 mutants and dia2Δ (which already strongly support the in vivo relevance of our in vitro experiments), we hope that the reviewer will agree that these experiments fall beyond the scope of a reasonable revision period for this current manuscript.

- An in vivo assessment of the termination function of Rrm3 mutations identified here may also help address why mutation of the GINS interaction motif in Rrm3 does not affect the replication termination function of Rrm3 in vitro yet exhibits a synthetic growth defect with DIA2 deletion in cells. Perhaps conditions are simply more stringent in vivo than in vitro and loss of the GINS interaction will exhibit a termination defect in vivo? This is relevant as the authors propose a similar or identical mode of action for Rrm3 at replisomes in both scenarios.

Disruption of the Rrm3-Dpb2 interaction (Rrm3Δ86-110) blocks Rrm3 recruitment to CMGE (Fig. 4D) and replication termination (Fig. 5B) in vitro,

and causes profound synthetic sickness in *dia2Δ* cells (Fig. 5D). In contrast, mutation of the GINS interaction motif in Rrm3 (Rrm3-2E) only partially reduces Rrm3 binding to CMGE in vitro (Fig. 4D), and, accordingly, has a less penetrant phenotype than *rrm3Δ86-110*, when combined with *dia2Δ* in yeast cells (compare Fig. 5C and 5D). Thus, overall, the in vitro and in vivo data for our Rrm3 mutants correlate very well with one another. However, the reviewer is right to highlight the lack of in vitro termination defect with Rrm3-2E (Fig. 5A). The underlying reason for this apparent discrepancy is still something that we do not fully understand. We agree with the reviewer that a plausible explanation for this difference is that the in vitro DNA replication experiments, which are necessarily performed in a buffer with low salt concentrations (100 mM KOAc), might simply be more permissive than the situation in cells, allowing Rrm3 to function at reconstituted replication forks, even in the absence of the Rrm3-Sld5 interaction. Unfortunately, despite our best efforts (as discussed above), we have been unable to take this issue further using cellular replication termination assays.

- The functional significance of the reported RTEL1 interactions with human CMG and Pol-epsilon remains untested altogether.

The primary conclusion relating to the RTEL1 experiments presented in our manuscript is that yeast Rrm3 and metazoan RTEL1 interact with the eukaryotic CMGE complex in a highly similar manner, indicative of convergent evolution. The AlphaFold modelling and protein-protein interaction studies presented in Fig. 6-7 support this conclusion. However, we accept that our data stops short of an assessment of the functional significance of CMGE-binding for RTEL1 function during DNA replication.

We agree that functional experiments with our new RTEL1 mutants represent a very interesting avenue for future investigation. One way to assess this would be via a functional in vitro biochemical assay that recapitulates human RTEL1 activity at reconstituted DNA replication forks. Unfortunately, however, the best currently available in vitro DNA replication system with human proteins (PMID: 35585232) involves the assembly of a single replisome at one end of a linear DNA template, and therefore does not recapitulate RTEL1-dependent DNA replication termination. Reconstituting replication termination as well as the other genome replication events that require RTEL1 (e.g. telomere replication, DNA-protein crosslink bypass) is a very interesting but longer term aim, that will require substantial development of this in vitro system. Alternatively, to characterise the RTEL1-CMGE interaction mutants in vivo would require CRISPR-modified RTEL1 mutant human cell lines, which would take some months to generate and characterise. For these reasons, we hope that the reviewer will agree that, whilst interesting, such experiments are beyond the scope of our current manuscript.

Additional points:

- DNA unwinding (helicase) assays and in vitro termination assays should be

performed in multiple repeats and the data quantified and plotted to allow assessment of the experimental variability and statistical significance of the observed effects.

This information was provided in our original manuscript for the in vitro replication termination assays in Fig. 2F and 5A-B, in Fig. EV2B (now Fig. EV2C) and EV4A-B, respectively. Stimulated by the reviewer's comment (as well as those from reviewer 1), we have also now included new experiments in our revised manuscript, involving three repeats of the helicase assays in Fig. EV1B (see Fig. EV1C for quantification) and 2C (see Fig. EV2B for quantification).

- A number of additional factors have been reported previously to mediate the recruitment of RTEL1 to sites of DNA replication, including TRF2 (PMID: 29727617), Poldip3 (PMCID: PMC7397856) and SLX4 (PMID 32398829). These reports should be discussed in light of the direct RTEL1-CMGE interaction proposed here.

We thank the reviewer for this suggestion, and agree that understanding the relative contributions of these various interactions to RTEL1 function at protein barriers, DNA-protein crosslinks, replication termination sites, telomeres etc. is an interesting issue. We have now mentioned these interactors in lines 433-435 of our revised manuscript.

- Figure EV2A: The data suggest that Pif1 is more proficient in mediating replication termination than Rrm3. Does Pif1 contain a disordered domain with predicted SLIMs similar to Rrm3? Or is it possible that accessory helicases can utilize distinct modes of coordination with the replisome as opposed to only the conserved interaction proposed here?

The reviewer raises an interesting point and is correct to note that, in our in vitro experiments, budding yeast Pif1 is routinely more proficient than Rrm3 in supporting replication termination. One likely explanation for this difference relates to the different behaviours of Rrm3 and Pif1 in their purified forms. Pif1 is easily purified in large quantities, and remains soluble across a range of buffer conditions. In contrast, Rrm3 is notoriously difficult to purify (to our knowledge, only one other group has ever been able to obtain full-length Rrm3 in an active form, PMID: 38267580), and readily forms insoluble aggregates under lower salt concentrations, which may partially compromise its activity in the in vitro DNA replication system, compared with Pif1. Importantly, although less active in supporting replication termination in vitro, Rrm3 is in fact more important than Pif1 for supporting termination in vivo (e.g. PMID: 30850330), which is one reason that we focussed our efforts on Rrm3 in this study.

The mechanism by which Pif1 is targeted to budding yeast replication forks is currently incompletely understood. Budding yeast Pif1 is composed of a central helicase domain that is flanked at both the N- and C-terminus by regions that are predicted to be disordered, similar to the Rrm3 IDR that we focussed on in this study. A very recent report (PMID: 38480846) indicates that Pif1 contains both RPA- and PCNA-interacting motifs in its N-terminal and C-

terminal IDRs, respectively, raising the possibility of an alternative mode of recruitment compared to Rrm3. However, whether or not additional replisome interactions (perhaps analogous to the CMGE-binding sites described in this manuscript) exist in Pif1 is an interesting issue for the future, and is the subject of ongoing experiments in our group.

Dear Dr. Deegan,

Thank you for submitting your manuscript for consideration by the EMBO Journal. It has now been seen by referees #1 and #2 whose comments are enclosed. As you will see, these referees find their concerns to be fully addressed and are in favour of publication. There remain only a couple of minor editorial requests before I can formally accept the manuscript.

- (1) Could you please compile Appendix Figures and Tables in one Appendix PDF and remove the figure legends from ms file and placed them below the corresponding figures
- (2) Could you please resize the SYNOPSIS IMAGE to 550x300-600 pixels (width x height, jpeg or png format).
- (3) Please define the asterisk in the legend of figure 2c; EV 1b.

Thank you for the opportunity to consider your work for publication. I look forward to your revision.

Yours sincerely,

Cornelius Schneider

Cornelius Schneider, PhD
Editor
The EMBO Journal
c.schneider@embojournal.org

We realize that it is difficult to revise to a specific deadline. In the interest of protecting the conceptual advance provided by the work, we recommend a revision within 3 months (22nd Sep 2024). Please discuss the revision progress ahead of this time with the editor if you require more time to complete the revisions. Use the link below to submit your revision:

Referee #1:

The edits adequately address all of the concerns that I previously raised.

Referee #2:

I didn't have any concerns with the original submission, and I don't have any concerns here. I once again recommend publication of this excellent work

All editorial and formatting issues were resolved by the authors.

Dear Dr. Deegan,

I am pleased to inform you that your manuscript has been accepted for publication in the EMBO Journal.

Yours sincerely,

Cornelius Schneider, PhD
Editor
The EMBO Journal
c.schneider@embojournal.org
